

# MicroHH 1.0: a computational fluid dynamics code for direct numerical simulation and large-eddy simulation of atmospheric boundary layer flows

Chiel C. van Heerwaarden[1,2], Bart J. H. van Stratum[1,2], Thijs Heus[3], Jeremy A. Gibbs[4], Evgeni Fedorovich[5], and Juan-Pedro Mellado[2]

[1]Meteorology and Air Quality Group, Wageningen University, Wageningen, The Netherlands
[2]Max Planck Institute for Meteorology, Hamburg, Germany
[3]Cleveland State University, Cleveland, OH, USA
[4]Department of Mechanical Engineering, University of Utah, UT, USA
[5]University of Oklahoma, Norman, OK, USA

*Correspondence to:* Chiel van Heerwaarden
(chiel.vanheerwaarden@wur.nl)

**Abstract.** This paper describes MicroHH 1.0, a new and open source (www.microhh.org) computational fluid dynamics code for the simulation of turbulent flows in the atmosphere. It is primarily made for direct numerical simulation, but also supports large-eddy simulation (LES). The paper covers the description of the governing equations, their numerical implementation, and the parametrizations included in the code. Furthermore, the paper presents the validation of the dynamical core in the form of

convergence and conservation tests, and comparison of simulations of channel flows and slope flows against well-established test cases. The full numerical model, including the associated parametrizations for LES, has been tested for a set of cases under stable and unstable conditions, under the Boussinesq and anelastic approximation, and with dry and moist convection under stationary and time-varying boundary conditions. The paper presents performance tests showing good scaling from 256 to 32,768 processes. The Graphical Processing Unit-enabled version of the code reaches speedups of more than an order of

magnitude with respect to the conventional code for a variety of cases.

## 1  Introduction

In this paper we present a description of MicroHH 1.0, a new Computational Fluid Dynamics code for the simulation of turbulent flows, with a focus on those in the atmosphere. MicroHH is designed for the direct numerical simulation (DNS) technique, but also supports the large-eddy simulation (LES) technique. Its applications range from neutral channel flows to

cloudy atmospheric boundary layers in large domains. MicroHH is written in C++ and the Graphical Processing Units-enabled parts of the code in CUDA. The simulation algorithms have been designed and are written from scratch with the goal to create a fast and highly parallel code that is able to run on machines with more than 10,000 cores. This is a key requirement for the code to be able to perform DNS at very high Reynolds numbers, or to do LES in domains that approach the synoptic scales. We decided to start from scratch, in order to be able to use C++ and its extensive possibilities in object oriented- and



metaprogramming. Furthermore, the implementation of a dynamical core that is fully fourth-order in space, which is very beneficial for DNS, but to retain the option to switch to second-order accuracy for LES, required a new code design.

Even though we started from scratch, many of the ideas are the results of our experiences with order codes. Here, DALES (Heus et al., 2010), UCLA-LES (Stevens et al., 2005), and PALM (Maronga et al., 2015), deserve a reference as MicroHH
could not have been possible without those.

This paper is built up as following: in Sect. 2, we provide a full description of the governing equations of the dynamical core, and their numerical implementation is discussed in Sect. 3. Subsequently, in Sect. 4 we present the parameterizations and their underlying assumptions. Section 6 discusses the technical details of the code. This is followed by a series of model tests on the validity and accuracy of the dynamical core in Sect. 8, and a series of more applied atmospheric flow cases based on
previous studies (Sect. 9). Hereafter, the parallel performance is evaluated (Sect. 10). After the future plans (Sect. 11) and the concluding remarks (Sect. 12), there is a short description where to get MicroHH, and where to find its tutorials and a selection of visualisations (Sect. 13).

## 2    Dynamical core: governing equations

The dynamical core of MicroHH solves the conservation equations of mass, momentum, and energy under the anelastic ap-
proximation (Bannon, 1996). Under this approximation, the state variables density, pressure, and temperature are described as small fluctuations (denoted with a prime in this paper) from corresponding vertical reference profiles (denoted with subscript zero) that are functions of height only. This form of the approximation directly simplifies to the Boussinesq approximation if the reference density $\rho_0$ is assumed to be constant with height $z$. To facilitate the subsequent discussion of the conservation equations, we define the scale height for density $H_\rho$ based on the reference density profile

$$20 \quad H_\rho \quad \equiv \quad \left( \frac{1}{\rho_0} \frac{d\rho_0}{dz} \right)^{-1}. \tag{1}$$

### 2.1    Conservation of mass

The conservation of mass is formulated using Einstein summation as

$$\frac{\partial \rho_0 u_i}{\partial x_i} \quad = \quad \rho_0 \frac{\partial u_i}{\partial x_i} + \rho_0 w H_\rho^{-1} = 0, \tag{2}$$

where $u_i$ is the velocity vector $(u, v, w)$ and $x_i$ is the position vector $(x, y, z)$. Under the Boussinesq approximation $(H_\rho \to \infty)$,
Eq. 2 simplifies to conservation of volume

$$\frac{\partial u_i}{\partial x_i} \quad = \quad 0. \tag{3}$$





## 2.2 Conservation of momentum and the equation of state

The momentum equation is written in the flux form, in order to assure the best possible mass and momentum conservation. The hydrostatic balance $dp_0/dz = -\rho_0 g$ has been subtracted to arrive at the perturbation form

$$
\frac{\partial u_i}{\partial t} = -\frac{1}{\rho_0}\frac{\partial \rho_0 u_i u_j}{\partial x_j} - \frac{\partial}{\partial x_i}\left(\frac{p'}{\rho_0}\right)
$$

$$
+ \delta_{i3}g\frac{\theta_v'}{\theta_{v0}} + \nu\frac{\partial^2 u_i}{\partial x_j^2} + F_i, \tag{4}
$$

where $p'$ is the perturbation pressure, $\delta$ is the Kronecker delta, $g$ is the gravity acceleration, $\theta_v'$ is the perturbation virtual potential temperature, $\theta_{v0}$ the reference virtual potential temperature, $\nu$ the kinematic viscosity, and vector $F_i$ represents external forces resulting from parameterizations or large-scale forcings.

The corresponding equation of state is (see Bannon (1996) for its derivation)

$$
\frac{\theta_v'}{\theta_{v0}} = \frac{p'}{\rho_0 g H_\rho} - \frac{\rho'}{\rho_0}. \tag{5}
$$

Under the Boussinesq approximation, the two equations simplify to

$$
\frac{\partial u_i}{\partial t} = -\frac{\partial u_i u_j}{\partial x_j} - \frac{1}{\rho_0}\frac{\partial p'}{\partial x_i}
$$

$$
+ \delta_{i3}g\frac{\theta_v'}{\theta_{v0}} + \nu\frac{\partial^2 u_i}{\partial x_j^2} + F_i, \tag{6}
$$

$$
\frac{\theta_v'}{\theta_{v0}} = -\frac{\rho'}{\rho_0}. \tag{7}
$$

## 2.3 Pressure equation

The equation to acquire the pressure is diagnostic, because density fluctuations are neglected in the mass conservation equation under the anelastic approximation (Eq. 2). To simplify the notation, we define a function $f(u_i)$ that contains all right-hand side terms of Eq. 4, except the pressure gradient. To arrive at the equation that allows us to solve for the pressure, we multiply the equation with the base density $\rho_0$ and take its divergence. Conservation of mass ensures that the tendency term vanishes, and an elliptic equation for pressure remains

$$
\frac{\partial}{\partial x_i}\left[\rho_0\frac{\partial}{\partial x_i}\left(\frac{p'}{\rho_0}\right)\right] = \frac{\partial \rho_0 f(u_i)}{\partial x_i}. \tag{8}
$$

Under the Boussinesq approximation the equation simplifies to

$$
\frac{\partial^2}{\partial x_i^2}\left(\frac{p'}{\rho_0}\right) = \frac{\partial f(u_i)}{\partial x_i}. \tag{9}
$$

In Sect. 3 we explain how these equations are solved.

## 2.4 Conservation of an arbitrary scalar

The conservation equation of an arbitrary scalar $\phi$ is written in flux form

$$
\frac{\partial \phi}{\partial t} = -\frac{1}{\rho_0}\frac{\partial \rho_0 u_j \phi}{\partial x_j} + \kappa_\phi\frac{\partial^2 \phi}{\partial x_j^2} + S_\phi, \tag{10}
$$



where $\kappa_\phi$ is the diffusivity of the scalar, and $S_\phi$ represents sources and sinks of the variable.

## 2.5 Conservation of energy

MicroHH provides multiple options for the energy conservation equation. The conservation equation for potential temperature for dry dynamics $\theta$ can be written as

$$\frac{\partial \theta}{\partial t} \;=\; -\frac{1}{\rho_0}\frac{\partial \rho_0 u_j \theta}{\partial x_j} + \kappa_\theta \frac{\partial^2 \theta}{\partial x_j^2} + \frac{\theta_0}{\rho_0 c_p T_0} Q, \tag{11}$$

where $\kappa_\theta$ is the thermal diffusivity for heat, and $Q$ represents sources and sinks of heat. A second option for moist dynamics is available. This has an identical conservation equation, but with liquid water potential temperature $\theta_l$ (moist dynamics), rather than $\theta$ as the conserved variable (see Sect. 3.8 for details).

A third, more simplified mode, is available for dry dynamics under the Boussinesq approximation. Here, the equation of state can be eliminated and the conservation of momentum and energy can be written using buoyancy $b \equiv (g/\rho_0)\rho'$ as

$$\frac{\partial u_i}{\partial t} + \frac{\partial u_i u_j}{\partial x_j} \;=\; -\frac{1}{\rho_0}\frac{\partial p'}{\partial x_i} + \delta_{i3} b + \nu \frac{\partial^2 u_i}{\partial x_j^2}, \tag{12}$$

$$\frac{\partial b}{\partial t} + \frac{\partial b u_j}{\partial x_j} \;=\; \kappa_b \frac{\partial^2 b}{\partial x_j^2}. \tag{13}$$

with $\kappa_b$ being the diffusivity for buoyancy.

With a slight modification to the previous set of equations, it is possible to study slope flows in periodic domains. If we introduce a slope $\alpha$ (positive anticlockwise) in the $x$-direction, take the proper gravity vector, and subtract the background buoyancy profile $N^2 z$ from the buoyancy value, the set of Eqs. 12 and 13 becomes

$$\frac{\partial u}{\partial t} + \frac{\partial u_j u}{\partial x_j} \;=\; -\frac{1}{\rho_0}\frac{\partial p'}{\partial x} + \sin(\alpha)b + \nu \frac{\partial^2 u}{\partial x_j^2}, \tag{14}$$

$$\frac{\partial w}{\partial t} + \frac{\partial u_j w}{\partial x_j} \;=\; -\frac{1}{\rho_0}\frac{\partial p'}{\partial z} + \cos(\alpha)b + \nu \frac{\partial^2 w}{\partial x_j^2}, \tag{15}$$

$$\frac{\partial b}{\partial t} + \frac{\partial b u_j}{\partial x_j} \;=\; \kappa_b \frac{\partial^2 b}{\partial x_j^2} - (u\sin(\alpha) + w\cos(\alpha))N^2 \tag{16}$$

where the evolution equation of $v$ is omitted as it contains no changes.

## 3 Dynamical core: numerical implementation

### 3.1 Time integration

The prognostic equations are solved using low-storage Runge-Kutta time integration schemes. Such schemes require two fields per variable: one that contains the actual value, which we denote with $\phi$ in this section, and one that represents the tendencies, denoted with $\delta\phi$. The code provides two options: a three-stage third-order scheme (Williamson, 1980) and a five-stage fourth-order scheme (Carpenter and Kennedy, 1994). Both can be written in the same generic form in semi-discrete formulation



as

$$(\delta\phi)_n \quad = \quad f(\phi_n) + a_n (\delta\phi)_{n-1} \tag{17}$$

$$\phi_{n+1} \quad = \quad \phi_n + b_n \Delta t (\delta\phi)_n, \tag{18}$$

where $f$ is a function that represents the computation of all right-hand side terms, $a_n$ and $b_n$ are the coefficients for the Runge-

Kutta method at stage $n$, and $\Delta t$ is the time step. Expression $f(\phi_n)$ represents thus the actual tendency calculated using, for

instance, Eqs. 4 or 10, whereas $(\delta\phi)_n$ is a composite of the actual tendency and those from the previous stages. In low-storage

form, the tendencies of the previous stage $(\delta\phi)_{n-1}$ are retained and multiplied with $a_n$ at the beginning of a stage, except for

the first stage, where $a_1 = 0$.

For the third-order scheme the vectors $a_n$ and $b_n$ are

$$a_n \quad = \quad \left\{ 0, -\frac{5}{9}, -\frac{153}{128} \right\}, \tag{19}$$

$$b_n \quad = \quad \left\{ \frac{1}{3}, \frac{15}{16}, \frac{8}{15} \right\}. \tag{20}$$

For the fourth-order scheme the vectors $a$ and $b$ are

$$a_n \quad = \quad \left\{ 0, -\frac{567301805773}{1357537059087}, -\frac{2404267990393}{2016746695238}, \right.$$
$$\left. -\frac{3550918686646}{2091501179385}, -\frac{1275806237668}{842570457699} \right\} \tag{21}$$

$$b_n \quad = \quad \left\{ \frac{1432997174477}{9575080441755}, \frac{5161836677717}{13612068292357}, \frac{1720146321549}{2090206949498}, \right.$$
$$\left. \frac{3134564353537}{4481467310338}, \frac{2277821191437}{14882151754819} \right\} \tag{22}$$

The reduced truncation error of the fourth-order scheme makes the scheme preferable over the third-order scheme under

many conditions (see Sect. 8.2). The code can be run with a fixed $\Delta t$, as well as an adaptive time step based on the local flow

velocities.

## 3.2   Grid

MicroHH is discretized on a staggered Arakawa C-grid, where the scalars are located in the center of a grid cell and the three

velocity components at the faces.

The code can work with stretched grids in the wall-bounded dimension. The grid is initialized from a vertical profile that

contains the heights of the cell centres. The locations of the faces are determined consistently with the spatial order of the

interpolations that are described in the next section.

There is the option to apply a uniform translation velocity to the grid, thus to let the grid move with the flow. This so-called

Galilean transformation is allowed as the Navier-Stokes equations are invariant under translation. It has the potential to allow

for larger time steps and to increase the accuracy of simulations.





### 3.3 Building blocks of the spatial discretization

The spatial operators are based on finite differences. The code supports second-order and fourth-order accurate discretizations following Morinishi et al. (1998); Vasilyev (2000). From Taylor series, spatial operators can be derived that constitute the building blocks of more advanced operators, such as the advection and diffusion operators. In the following subsections we

describe the elementary operators and the composite operators that can be derived from them. We use only two dimensions for brevity.

We define two second-order interpolation operators, one with a small stencil and one with a wide stencil, as

$$\phi_{i,j} \approx \widehat{\phi}_{i,j}^{2x} \quad \equiv \quad \frac{\phi_{i-\frac{1}{2},j} + \phi_{i+\frac{1}{2},j}}{2}, \tag{23}$$

$$\phi_{i,j} \approx \widehat{\phi}_{i,j}^{2xL} \quad \equiv \quad \frac{\phi_{i-\frac{3}{2},j} + \phi_{i+\frac{3}{2},j}}{2}, \tag{24}$$

Interpolations are marked with a hat. The superscript indicates the spatial order $(2)$, and the direction $(x)$ and has an extra qualifier $L$ when it is taken using the wide stencil. The subscript indicates the position on the grid $(i,j)$.

The gradient operators, denoted with letter $\delta$, are defined in a similar way

$$\left.\frac{\partial \phi}{\partial x}\right|_{i,j} \approx \delta^{2x}\left(\phi\right)_{i,j} \quad \equiv \quad \frac{\phi_{i+\frac{1}{2},j} - \phi_{i-\frac{1}{2},j}}{x_{i+\frac{1}{2}} - x_{i-\frac{1}{2}}} \tag{25}$$

$$\left.\frac{\partial \phi}{\partial x}\right|_{i,j} \approx \delta^{2xL}\left(\phi\right)_{i,j} \quad \equiv \quad \frac{\phi_{i+\frac{3}{2},j} - \phi_{i-\frac{3}{2},j}}{x_{i+\frac{3}{2}} - x_{i-\frac{3}{2}}} \tag{26}$$

We use the Einstein summation in the operators. For instance, the divergence of vector $u_i|_{i,j}$ can be written as $\delta^{2x_i}\left(u_i\right)_{i,j}$.

The fourth-order operators, written down in the same notation, are defined as

$$\phi_{i,j} \approx \widehat{\phi}_{i,j}^{4x} \equiv \frac{-\phi_{i-\frac{3}{2},j} + 9\phi_{i-\frac{1}{2},j} + 9\phi_{i+\frac{1}{2},j} - \phi_{i+\frac{3}{2},j}}{16}. \tag{27}$$

The biased version of this operator (subscript $b$) can be applied in the vicinity of the boundaries

$$\phi_{i,j} \approx \widehat{\phi}_{i,j}^{4xb} \equiv \frac{5\phi_{i-\frac{1}{2},j} + 15\phi_{i+\frac{1}{2},j} - 5\phi_{i+\frac{3}{2},j} + \phi_{i+\frac{5}{2},j}}{16}. \tag{28}$$

Note that we only write down the bottom boundary for brevity.

The centered and biased fourth-order gradient operators are

$$\left.\frac{\partial \phi}{\partial x}\right|_{i,j} \quad \approx \quad \delta^{4x}\left(\phi\right)_{i,j}$$

$$\equiv \quad \frac{\phi_{i-\frac{3}{2},j} - 27\phi_{i-\frac{1}{2},j} + 27\phi_{i+\frac{1}{2},j} - \phi_{i+\frac{3}{2},j}}{x_{i-\frac{3}{2}} - 27x_{i-\frac{1}{2}} + 27x_{i+\frac{1}{2}} - x_{i+\frac{3}{2}}}, \tag{29}$$

and

$$\left.\frac{\partial \phi}{\partial x}\right|_{i,j} \quad \approx \quad \delta^{4xb}\left(\phi\right)_{i,j}$$

$$\equiv \quad \frac{-23\phi_{i-\frac{1}{2},j} + 21\phi_{i+\frac{1}{2},j} + 3\phi_{i+\frac{3}{2},j} - \phi_{i+\frac{5}{2},j}}{-23x_{i-\frac{1}{2}} + 21x_{i+\frac{1}{2}} + 3x_{i+\frac{3}{2}} - x_{i+\frac{5}{2}}} \tag{30}$$





### 3.4 Advection

We use the previously introduced notation to describe the more complex operators and expand them for illustration. The advection term is discretized in the flux form, where $\phi$ is an arbitrary scalar located in the center of the grid cell. In the second-order case, this gives the following discretization:

$$
\begin{aligned}
\left.\frac{\partial u\phi}{\partial x}\right|_{i,j} + \left.\frac{\partial v\phi}{\partial y}\right|_{i,j} &\approx \delta^{2x}\left(u\widehat{\phi}^{2x}\right)_{i,j} + \delta^{2y}\left(v\widehat{\phi}^{2y}\right)_{i,j} \\
&= \frac{u_{i+\frac{1}{2},j}\widehat{\phi}^{2x}_{i+\frac{1}{2},j} - u_{i-\frac{1}{2},j}\widehat{\phi}^{2x}_{i-\frac{1}{2},j}}{x_{i+\frac{1}{2}} - x_{i-\frac{1}{2}}} \\
&+ \frac{v_{i,j+\frac{1}{2}}\widehat{\phi}^{2y}_{i,j+\frac{1}{2}} - v_{i,j-\frac{1}{2}}\widehat{\phi}^{2y}_{i,j-\frac{1}{2}}}{y_{j+\frac{1}{2}} - y_{j-\frac{1}{2}}}
\end{aligned}
\tag{31}
$$

The discretization of the advection of the velocity components (see Eqs. 4 and 6) involves extra interpolations as the following example illustrates:

$$
\begin{aligned}
\left.\frac{\partial vu}{\partial x}\right|_{i,j} &= \delta^{2x}\left(\widehat{v}^{2y}\widehat{u}^{2x}\right)_{i,j} \\
&= \frac{\widehat{v}^{2y}_{i+\frac{1}{2},j}\widehat{u}^{2x}_{i+\frac{1}{2},j} - \widehat{v}^{2y}_{i-\frac{1}{2},j}\widehat{u}^{2x}_{i-\frac{1}{2},j}}{x_{i+\frac{1}{2}} - x_{i-\frac{1}{2}}}
\end{aligned}
\tag{32}
$$

In the standard fourth-order scheme, the scalar advection in flux form is represented by

$$
\begin{aligned}
\left.\frac{\partial u\phi}{\partial x}\right|_{i,j} &\approx \delta^{4x}\left(u\widehat{\phi}^{4x}\right)_{i,j} \\
&= \Big(u_{i-\frac{3}{2},j}\widehat{\phi}^{4x}_{i-\frac{3}{2},j} - 27u_{i-\frac{1}{2},j}\widehat{\phi}^{4x}_{i-\frac{1}{2},j} \\
&\quad + 27u_{i+\frac{1}{2},j}\widehat{\phi}^{4x}_{i+\frac{1}{2},j} - u_{i+\frac{3}{2},j}\widehat{\phi}^{4x}_{i+\frac{3}{2},j}\Big) \\
&\quad \Big/ \Big(x_{i-\frac{3}{2}} - 27x_{i-\frac{1}{2}} + 27x_{i+\frac{1}{2}} - x_{i+\frac{3}{2}}\Big).
\end{aligned}
\tag{33}
$$

Hereafter, we assume that operator notation is clear and only expand it where necessary.

MicroHH has a fully energy-conserving fourth-order advection scheme (Morinishi et al., 1998) available. This scheme is constructed by interpolation of two energy-conserving second-order schemes to eliminate the second-order error as illustrated below

$$
\left.\frac{\partial u\phi}{\partial x}\right|_{i,j} \approx \frac{9}{8}\delta^{2x}\left(u\widehat{\phi}^{2x}\right)_{i,j} - \frac{1}{8}\delta^{2xL}\left(u\widehat{\phi}^{2xL}\right)_{i,j}
\tag{34}
$$

Velocity interpolations, such as those in Eq. 32, still need to be performed with fourth-order accuracy (Eq. 27) in order to be fourth-order accurate (see Morinishi et al. (1998) for details). The expression

$$
\left.\frac{\partial vu}{\partial x}\right|_{i,j} \approx \frac{9}{8}\delta^{2x}\left(\widehat{v}^{4y}\widehat{u}^{2x}\right)_{i,j} - \frac{1}{8}\delta^{2xL}\left(\widehat{v}^{4y}\widehat{u}^{2xL}\right)_{i,j}
\tag{35}
$$

includes, for instance, a combination of second- and fourth-order interpolations.

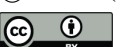



To increase the overall accuracy of the second-order advection operator, there is an option available to only increase the interpolation part to fourth order

$$\left.\frac{\partial u\phi}{\partial x}\right|_{i,j} \approx \delta^{2x}\left(u\widehat{\phi}^{4x}\right)_{i,j}. \tag{36}$$

### 3.5 Diffusion

We apply a discretization for diffusion that can be written as the divergence of a gradient, using the building blocks defined earlier in this section. As this operator is identical in all directions, we present it in one direction only

$$\left.\kappa_\phi\frac{\partial^2\phi}{\partial x^2}\right|_{i,j} \approx \kappa_\phi\delta^{2x}\left(\delta^{2x}\left(\phi\right)\right)_{i,j} \tag{37}$$

$$\left.\kappa_\phi\frac{\partial^2\phi}{\partial x^2}\right|_{i,j} \approx \kappa_\phi\delta^{4x}\left(\delta^{4x}\left(\phi\right)\right)_{i,j} \tag{38}$$

On an equidistant grid, this provides the well-known second-order accurate operator for the second derivative

$$\kappa_\phi\delta^{2x}\left(\delta^{2x}\left(\phi\right)\right)_{i,j} = \kappa_\phi\frac{\phi_{i-1,j} - 2\phi_{i,j} + \phi_{i+1,j}}{\left(\Delta x\right)^2}, \tag{39}$$

where $\Delta x$ is the uniform grid spacing.

For the fourth-order accurate operator, a seven-point stencil is used:

$$\begin{aligned}
&\kappa_\phi\delta^{4x}\left(\delta^{4x}\left(\phi\right)\right)_{i,j}\\
&= \frac{\kappa_\phi}{576\left(\Delta x\right)^2}\left(\phi_{i-3,j} - 54\phi_{i-2,j} + 783\phi_{i-1,j}\right.\\
&\left. -1460\phi_{i,j} + 783\phi_{i+1,j} - 54\phi_{i+2,j} + \phi_{i+3,j}\right).
\end{aligned} \tag{40}$$

The seven point wide stencil and its properties has been discussed in detail in Castillo et al. (1995).

### 3.6 Pressure

Eqs. 8 and 9 are solved following the method of Chorin (1968). This is a fractional step method that first computes intermediate values of the velocity components for the next time step, based on all right hand side terms of the momentum conservation equation Eq. 4

$$\left.\widetilde{u}_i\right|_{i,j,k}^{t+1} = \left.u_i\right|_{i,j,k}^t + \Delta t\left.f_i\right|_{i,j,k}^t, \tag{41}$$

with the intermediate velocity components denoted with a tilde.

The velocity values at the next time step can be computed as soon as the pressure is known, using

$$\left.u_i\right|_{i,j,k}^{t+1} = \left.\widetilde{u}_i\right|_{i,j,k}^{t+1} - \Delta t\,\delta^{nx_i}\left.\left(\frac{p}{\rho_0}\right)\right|_{i,j,k}^t. \tag{42}$$

In order to compute the pressure, we multiply the previous equation with the reference density and take its gradient, arriving at

$$\begin{aligned}
\left.\delta^{nx_i}\left(\rho_0 u_i\right)\right|_{i,j,k}^{t+1} &= \left.\delta^{nx_i}\left(\rho_0\widetilde{u}_i\right)\right|_{i,j,k}^{t+1}\\
&- \Delta t\,\delta^{nx_i}\left.\left[\rho_0\delta^{nx_i}\left(\frac{p}{\rho_0}\right)\right]\right|_{i,j,k}^t,
\end{aligned} \tag{43}$$





where $n$ indicates the spatial order, and the subscript $i$ in superscript $x_i$ indicates that $\delta^{nx_i}$ is a divergence operator. The left hand side equals zero due to mass conservation at the next time step (Eq. 2). The resulting equation is the Poisson equation that is the discrete equivalent of Eq. 8. Rewriting this equation leads to

$$\frac{\delta^{nx_i}\left(\rho_0\widetilde{u_i}\right)\big|_{i,j,k}^{t+1}}{\Delta t} \quad = \quad \delta^{nx_i}\left[\rho_0\delta^{nx_i}\left(\frac{p}{\rho_0}\right)\right]\Bigg|_{i,j,k}^{t}. \tag{44}$$

To simplify the notation, we denote the left-hand side term as $\psi$ and the $p/\rho_0$ term on the right hand side as $\pi$. Solving a Poisson equation is a global operation. Because the computed fields are periodic in the horizontal directions on an equidistant grid, and a Poisson equation is linear, we can perform a Fourier transform in the two horizontal directions

$$\widehat{\psi}_{l,m,k} = -k_{*n}^2\widehat{\pi}_{l,m,k} - l_{*n}^2\widehat{\pi}_{l,m,k} + \delta^{nz}\left[\rho_0\delta^{nz}\left(\widehat{\pi}\right)\right]_{l,m,k}, \tag{45}$$

where Fourier transformed variables are denoted with a hat, the spatial order of the operation with $n$, and the wave numbers

in the two horizontal dimensions $x$ and $y$ are $l$ and $m$ respectively. Variables $k_*^2$ and $l_*^2$ are the squares of the modified wave numbers

$$-k_{*2}^2 \quad \equiv \quad 2\frac{\cos(k\Delta x)}{(\Delta x)^2} - \frac{2}{(\Delta x)^2} \tag{46}$$

and

$$-k_{*4}^2 \quad \equiv \quad 2\frac{\cos(3k\Delta x) - 54\cos(2k\Delta x) + 783\cos(k\Delta x)}{576\left(\Delta x\right)^2}$$

$$\quad - \quad \frac{1460}{576\left(\Delta x\right)^2}, \tag{47}$$

where the former is the modified wave number for the second-order accurate solver and the latter is the wave number for the fourth-order one. Both expressions satisfy the limit $\lim_{\Delta x\to 0} k_{*n}^2 = k^2$, where $n$ is the order of the scheme.

Solving Eq. 45 for $\widehat{\pi}$ requires solving a banded matrix, which is tridiagonal for the second-order solver and hepta-diagonal for the fourth-order solver. For this, a standard Thomas algorithm is used. After the pressure is acquired, inverse Fourier

transforms are applied and subsequently the pressure gradient term (see Eqs. 4 and 6) is computed for all three components of the velocity tendency. Note that the computation of the corrected velocity components does not require a boundary condition for pressure (see Vreman (2014) for details).

### 3.7   Boundary conditions

The lateral boundaries in MicroHH are periodic. The bottom and top boundary conditions can be formulated in their most

general form as the Robin boundary condition

$$a\phi_s + b\left.\frac{\partial\phi}{\partial z}\right|_s = c, \tag{48}$$

with $a$, $b$ and $c$ as constants. This gives the Dirichlet boundary condition when $a=1$, $b=0$, and the Neumann boundary condition when $a=0$, $b=1$.





MicroHH makes use of ghost cells in order to avoid the need of biased schemes for single interpolation or gradient operators near the wall. The values at the ghost cells are derived making use of the boundary conditions following Morinishi et al. (1998). The ghost cells for the Dirichlet boundary conditions in the second-order accurate discretization are

$$\phi_{-\frac{1}{2}} = 2c - \phi_{\frac{1}{2}}, \tag{49}$$

whereas those for the Neumann boundary condition are

$$\phi_{-\frac{1}{2}} = -c\left(-z_{-\frac{1}{2}} + z_{\frac{1}{2}}\right) + \phi_{\frac{1}{2}}. \tag{50}$$

In case of the fourth-order scheme, we have two ghost cells, and therefore a second boundary condition is required. Here, we set the third derivative equal to zero following (Morinishi et al., 1998). For the Dirichlet boundary condition we then acquire the following expressions for the ghost cells

$$\phi_{-\frac{1}{2}} = \frac{8c - 6\phi_{\frac{1}{2}} + \phi_{\frac{3}{2}}}{3}, \tag{51}$$

$$\phi_{-\frac{3}{2}} = 8c - 6\phi_{\frac{1}{2}} + \phi_{\frac{3}{2}}, \tag{52}$$

whereas in case of a Neumann boundary condition we find

$$\phi_{-\frac{1}{2}} = -c\frac{z_{i-\frac{3}{2}} - 27z_{i-\frac{1}{2}} + 27z_{i+\frac{1}{2}} - z_{i+\frac{3}{2}}}{24} + \phi_{\frac{1}{2}}, \tag{53}$$

$$\phi_{-\frac{3}{2}} = -3c\frac{z_{i-\frac{3}{2}} - 27z_{i-\frac{1}{2}} + 27z_{i+\frac{1}{2}} - z_{i+\frac{3}{2}}}{24} + \phi_{\frac{3}{2}}. \tag{54}$$

## 3.8 Thermodynamics

MicroHH supports the potential ($\theta$) and liquid water potential ($\theta_l$) temperature as thermodynamic variables (Sect. 2.5). The dry ($\theta$) and moist ($\theta_l$) thermodynamics are related through the use of a total specific humidity $q_t$, which is defined as the sum of the water vapour specific humidity ($q_v$) and the cloud liquid water specific humidity ($q_l$). In the absence of liquid water, $\theta_l = \theta$, in the presence of liquid water, the liquid water potential temperature is approximated as (Betts, 1973)

$$\theta_l \approx \theta - \frac{L_v}{c_p \Pi} q_l, \tag{55}$$

where $L_v$ is the latent heat of vaporization, $c_p$ the specific heat of dry air at constant pressure, and $\Pi$ is the Exner function

$$\Pi = \left(\frac{p}{p_{00}}\right)^{R_d/c_p}, \tag{56}$$

where $p$ is the hydrostatic pressure, $p_{00}$ a constant reference pressure, and $R_d$ the gas constant for dry air. The cloud liquid water content is calculated as

$$q_l = \max(0,\, q_t - q_s), \tag{57}$$



where $q_s$ is the saturation specific humidity

$$q_s = \frac{\epsilon\, e_s}{p - (1-\epsilon)\, e_s},$$ (58)

with $\epsilon$ the ratio between the gas constant for dry air and the gas constant for water vapour ($R_d/R_v$), and $e_s$ the saturation vapor pressure. The latter is approximated using a $10^{\text{th}}$ order Taylor expansion at $T = 0$ degree Celsius of the Arden Buck equation (Buck, 1981). $q_l$ is adjusted iteratively to arrive at a consistent state where $q_v = q_s$. Finally, the virtual potential temperature (Eq. 4) is defined in MicroHH as

$$\theta_v \equiv \theta\left(1 - \left[1 - \frac{R_v}{R_d}\right]q_t - \frac{R_v}{R_d}q_l\right)$$ (59)

The base state pressure and density are calculated assuming a hydrostatic equilibrium: $dp_0 = -\rho_0 g dz$, with the density defined as $\rho_0 = p_0/(R_d\, \Pi\, \theta_{v0})$. Integration with height results in

$$p_{0;k+1} = p_{0;k}\, \exp\left(\frac{-g(z_{k+1} - z_k)}{R_d\, \Pi\, \theta_{v0}}\right)$$ (60)

where $\theta_{v0}$ is the average virtual potential temperature between $z_k$ and $z_{k+1}$. This equation is applied from a given surface pressure to the model top, alternating the calculations at the full and half model levels. That is, given the full thermodynamic state (pressure and density) at a full level $k$, the thermodynamic state can be advanced from the half level $k - \frac{1}{2}$ to $k + \frac{1}{2}$. Using the newly calculated state at $k + \frac{1}{2}$, pressure and density at $k + 1$ can be calculated.

The base state density $\rho_0$ that is used in the dynamical core (Sect. 2) is calculated using the initial virtual potential temperature profile, and is not updated during the experiment. The density and hydrostatic pressure used in the moist thermodynamics can optionally be updated every time step, following the same procedure as explained in Boing (2014).

### 3.9 Rotation

The effects of a rotating reference can be included through the Coriolis force. MicroHH can run on an $f$-plane, where the related tendencies of the two horizontal velocity components are calculated as

$$\left.\frac{\partial u}{\partial t}\right|_{i,j,k,F_{\text{cor}}} = f_0 v_{i,j,k},$$ (61)

$$\left.\frac{\partial v}{\partial t}\right|_{i,j,k,F_{\text{cor}}} = -f_0 u_{i,j,k},$$ (62)

with $f_0$ as Coriolis parameter specified by the user.



## 4 Physical parameterizations

### 4.1 Subfilter-scale model for large-eddy simulation

With the governing equations described in Sect. 2 it is possible to resolve the flow down to the scales where molecular viscosity acts. In many applications, however, such simulations are too costly. In that case, one may opt for large-eddy simulation (LES),

5  where filtered equations are used to describe the largest scales of the flow, and the subfilter-scale motions are modeled. Filtering of the anelastic conservation of momentum equation (Eq. 4), with a tilde applied to denote filtered variables, leads to

$$
\frac{\partial \widetilde{u}_i}{\partial t} = -\frac{1}{\rho_0}\frac{\partial \rho_0 \widetilde{u}_i \widetilde{u}_j}{\partial x_j} - \frac{\partial \pi}{\partial x_i} - \frac{1}{\rho_0}\frac{\partial \rho_0 \tau_{ij}}{\partial x_j}
$$
$$
+ \quad \delta_{i3} g \frac{\widetilde{\theta'_v}}{\theta_{v0}} + F_i. \tag{63}
$$

In this equation, a tensor $\tau_{ij}$ is defined as

$$
\tau_{ij} \equiv \widetilde{u_i u_j} - \widetilde{u}_i \widetilde{u}_j - \frac{1}{3}\left(\widetilde{u_i u_i} - \widetilde{u}_i \widetilde{u}_i\right). \tag{64}
$$

This is the anisotropic subfilter-scale kinematic momentum flux tensor. The isotropic part of the full momentum flux tensor has been added to the pressure, providing the modified pressure

$$
\pi \equiv \frac{\widetilde{p}'}{\rho_0} + \frac{1}{3}\left(\widetilde{u_i u_i} - \widetilde{u}_i \widetilde{u}_i\right). \tag{65}
$$

As $\tau_{ij}$ contains the filtered product of unfiltered velocity components, this quantity needs to be parameterized. MicroHH uses

15  the Smagorinsky-Lilly (Lilly, 1968) model, in which $\tau_{ij}$ is modeled as

$$
\tau_{ij} = -K_m\left(\frac{\partial \widetilde{u}_i}{\partial x_j} + \frac{\partial \widetilde{u}_j}{\partial x_i}\right), \tag{66}
$$

with $K_m$ interpreted as the subfilter eddy-diffusivity. This quantity is modeled as

$$
K_m = \lambda^2 \left(2 S_{ij} S{ij}\right)^{\frac{1}{2}} \left(1 - \frac{N^2}{Pr_t S^2}\right)^{\frac{1}{2}}, \tag{67}
$$

and is proportional to the magnitude of $S$ of the strain tensor

$$
S_{ij} \equiv \frac{1}{2}\left(\frac{\partial \widetilde{u}_i}{\partial x_j} + \frac{\partial \widetilde{u}_j}{\partial x_i}\right). \tag{68}
$$

The subfilter eddy diffusivity thus takes into account the local stratification $N^2 \equiv (g/\theta_{v0})/(\partial \widetilde{\theta}_v/\partial z)$ and the turbulent Prandtl number $Pr_t$. The latter is set to $\frac{1}{3}$ by default, but can be overridden in the settings. The length scale $\lambda$ is the mixing length defined following Mason and Thomson (1992), as

$$
\frac{1}{\lambda^n} = \frac{1}{[\kappa (z + z_0)]^n} + \frac{1}{(c_s \Delta)^n}, \tag{69}
$$

25  which is an arbitrary ($n$ is a free parameter, set to 2 in MicroHH) matching function between the mixing length following wall scaling to the subfilter length scale (filter size) $\Delta \equiv (\Delta x \Delta y \Delta z)^{\frac{1}{3}}$, related to the grid spacing. In case of a high-Reynolds



number atmospheric LES with an unresolved near-wall flow, the vertical gradients of the horizontal velocity components $\partial \widetilde{u}_{i,j}/\partial z$ in the strain tensor are replaced with the theoretical gradients predicted from Monin-Obukhov similarity theory. Evaluation of these gradients is explained in detail in Section 4.2.

The same approach is followed for all scalars, including the thermodynamic variables discussed in Sect. 2.5:

$$\frac{\partial \widetilde{\phi}}{\partial t} = -\frac{1}{\rho_0}\frac{\partial \rho_0 \widetilde{u}_j \widetilde{\phi}}{\partial x_j} - \frac{1}{\rho_0}\frac{\partial \rho_0 R_{\phi,j}}{\partial x_j} + \widetilde{S}_\phi. \tag{70}$$

The term $R_{\phi,j}$ refers to the subfilter flux of $\widetilde{\phi}$ and is defined as

$$R_{\phi,j} = \widetilde{u_j \phi} - \widetilde{u}_j \widetilde{\phi}. \tag{71}$$

The subfilter-scale flux is parameterized in terms of the gradient

$$R_{\phi,j} = -\frac{K_m}{Pr_t}\frac{\partial \widetilde{\phi}}{\partial x_j}. \tag{72}$$

## 4.2 Surface model

MicroHH uses a surface model to compute the surface fluxes of the horizontal momentum components and the scalars (including thermodynamic variables) in flows over rough surfaces at high Reynolds numbers. This is a typical configuration for atmospheric flows. The surface model is entirely built on Monin-Obukhov Similarity Theory (MOST) (see Wyngaard (2010)) that relates surface fluxes of variables to their near-surface gradients using empirical functions that depend on the height of the first model level $z_1$ divided by the Obukhov length $L$ as an argument. Length $L$ is defined as

$$L \equiv -\frac{u_*^3}{\kappa B_0}, \tag{73}$$

where $u_*$ is the friction velocity, $\kappa$ is the Von Karman constant and $B_0$ is the surface buoyancy flux. $L$ represents the height at which the buoyancy production/destruction of turbulence kinetic energy equals the shear production.

Following MOST, the friction velocity $u_*$ and the momentum fluxes may be related to the near-surface wind gradient as

$$\frac{\kappa z_1}{u_*}\frac{\partial U}{\partial z} = -\frac{\kappa z_1 u_*}{\overline{u'w'}}\frac{\partial \widetilde{u}}{\partial z} = -\frac{\kappa z_1 u_*}{\overline{v'w'}}\frac{\partial \widetilde{v}}{\partial z} = \phi_m\left(\frac{z_1}{L}\right), \tag{74}$$

where $U$ is defined as $\sqrt{\widetilde{u}^2 + \widetilde{v}^2}$, and $\overline{u'w'}$ and $\overline{v'w'}$ as the surface momentum fluxes for the two wind components. These relationships can be integrated from the roughness length $z_{0m}$ to $z_1$ resulting in

$$u_* = f_m\left(U_1 - U_0\right), \tag{75}$$

$$\overline{u'w'} = -u_* f_m\left(\widetilde{u}_1 - \widetilde{u}_0\right), \tag{76}$$

$$\overline{v'w'} = -u_* f_m\left(\widetilde{v}_1 - \widetilde{v}_0\right), \tag{77}$$

with $f_m$ defined as:

$$f_m \equiv \frac{\kappa}{\ln\left(\frac{z_1}{z_{0m}}\right) - \Psi_m\left(\frac{z_1}{L}\right) + \Psi_m\left(\frac{z_{0m}}{L}\right)}, \tag{78}$$



with $\Psi_m$ desribed in Eqs. 83 and 85.

The same procedure for scalars is followed, with

$$\frac{\kappa z_1 u_*}{\overline{\phi' w'}} \frac{\partial \widetilde{\phi}}{\partial z} = \phi_h \left( \frac{z_1}{L} \right), \tag{79}$$

and in integrated form

$$\overline{\phi' w'} \;\; = \;\; u_* f_h \left( \widetilde{\phi}_1 - \widetilde{\phi}_0 \right), \tag{80}$$

with

$$f_h \;\; \equiv \;\; \frac{\kappa}{\ln \left( \dfrac{z_1}{z_{0h}} \right) - \Psi_h \left( \dfrac{z_1}{L} \right) + \Psi_h \left( \dfrac{z_{0h}}{L} \right)}, \tag{81}$$

with $\Psi_h$ desribed in Eqs. 83 and 85.

The functions $\phi_m$, $\phi_h$, $\Psi_m$, and $\Psi_h$ are empirical and depend on the static stability of the atmosphere. Under unstable conditions we follow (Wilson, 2001; Wyngaard, 2010)

$$\phi_{m,h} \;\; = \;\; \left( 1 + \gamma_{m,h} \, |\zeta|^{2/3} \right)^{-1/2}, \tag{82}$$

$$\Psi_{m,h} \;\; = \;\; 3 \ln \left( \frac{1 + \phi_{m,h}^{-1}}{2} \right), \tag{83}$$

where $\zeta$ is the ratio of a height and the Obukhov length $L$, $\gamma_m = 3.6$ and $\gamma_h = 7.9$. Under stable conditions we use (Högström, 1988; Wyngaard, 2010)

$$\phi_{m,h} \;\; = \;\; 1 + \lambda_{m,h} \zeta, \tag{84}$$

$$\Psi_{m,h} \;\; = \;\; -\lambda_{m,h} \zeta, \tag{85}$$

where $\lambda_m = 4.8$ and $\lambda_h = 7.8$.

With the equations above, the surface fluxes, surface values and near-surface gradients can be computed, but only if the Obukhov length $L$ is known. The surface model calculates the Obukhov length by relating the dimensionless parameter $z_1/L$ to a Richardson number. The employed formulation of the Richardson number depends on the chosen boundary condition in the model. Three possible options are available:

- fixed momentum fluxes and a fixed surface buoyancy flux. Both the friction velocity $u_*$ and the surface buoyancy flux $B_0$ are specified. Under these conditions we define the Richardson number $Ri_a$ equal to $z_1/L$; $L$ can be computed directly from the expression

$$Ri_a \equiv \frac{z_1}{L} = -\frac{\kappa z_1 B_0}{u_*^3}. \tag{86}$$

- a fixed horizontal velocity $U_0$ at the surface and a fixed surface buoyancy flux $B_0$. The friction velocity $u_*$ is unknown. Now, $L$ needs to be retrieved from the implicit relationship

$$Ri_b \equiv \frac{z_1}{L} f_m^3 = -\frac{\kappa z_1 B_0}{(U_1 - U_0)^3}. \tag{87}$$





– a fixed surface velocity $U_0$ and a fixed surface buoyancy $b_0$. With this boundary condition, the surface values of the horizontal velocities and the buoyancy are given, and both $u_*$ and the surface buoyancy flux $B_0$ are unknown. $L$ is then retrieved from

$$Ri_c \equiv \frac{z_1}{L} \frac{f_m^2}{f_h} = \frac{\kappa z_1 \left( \widetilde{b}_1 - \widetilde{b}_0 \right)}{\left( U_1 - U_0 \right)^2}. \tag{88}$$

In cases of the latter two options, a lookup table is created with $L$ as a function of Richardson number. The lookup table has $10^4$ entries, of which 90 percent is spaced uniformly between $z_1/L = -5$ to $5$. The remaining 10 percent are used to stretch the negative range up to $z_1/L = -10^4$ to allow for the correct free convection limit.

### 4.3 Large-scale forcings

#### 4.3.1 Pressure force

MicroHH provides two options to introduce a large-scale pressure force into the model. The first is to enforce a constant massflux through the domain in the streamwise direction. In this approach the desired large-scale velocity $U_f$ is set, and the corresponding pressure gradient is computed. We follow here the approach of van Reeuwijk (2007). In this approach, the $u$-component of the horizontal momentum equation (Eq. 4) is volume-averaged to acquire

$$\frac{\langle u \rangle^{n+1} - \langle u \rangle^n}{\Delta t} = \langle f_1 \rangle - \left\langle \frac{\partial}{\partial x} \left( \frac{p}{\rho_0} \right) \right\rangle + F_{p;ls} \tag{89}$$

where brackets indicate a volume average, $f_1$ contains all the righthand side terms of the $u$-component of the conservation of momentum, except for the dynamic pressure, which is contained in the second term. The large-scale pressure force $F_{p;ls}$, which is to be computed, is the last term. We can now set $\langle u \rangle^{n+1} = U_f$ to explicitly set the volume-averaged velocity in the next time step. Furthermore, the volume-averaged horizontal pressure gradient vanishes, because of the periodic boundary condition, which makes $F_{p;ls}$ the only unknown. The acquired pressure force will be added following

$$\left. \frac{\partial u}{\partial t} \right|_{i,j,k,F_{p;ls}} = F_{p;ls} \tag{90}$$

The second option is to enforce a large-scale pressure force through the geostrophic wind components $u_g$ and $v_g$, in combination with rotation, with the tendencies of the two horizontal velocity components calculated as

$$\left. \frac{\partial u}{\partial t} \right|_{i,j,k,F_{p;ls}} = -f_0 v_{g;k}, \tag{91}$$

$$\left. \frac{\partial v}{\partial t} \right|_{i,j,k,F_{p;ls}} = f_0 u_{g;k}, \tag{92}$$

where $u_{g;k}$ and $v_{g;k}$ as user-specified vertical profiles of geostrophic wind components.

#### 4.3.2 Large-scale sources and sinks

Large-scale sources and sinks, representing for instance large-scale adveciton or radiative cooling, can be prescribed for each variable separately. The user has to provide vertical profiles of large-scale tendencies $S_{\phi;ls}$ that are added to the total tendencies.





### 4.3.3 Large-scale vertical velocity

A second method of introducing large-scale thermodynamic effects is through the inclusion of a large-scale vertical velocity. In this case, each scalar gets an additional tendency term of the form

$$\frac{\partial \phi}{\partial t}\bigg|_{i,j,k,ls} = -w_{ls;k}\,\delta^{2x}\left(\langle\phi\rangle_k\right), \tag{93}$$

where $\langle\phi\rangle_k$ is the horizontally-averaged vertical profile at height $z_k$ for scalar $\phi$.

### 4.4 Buffer layer

MicroHH has the option to damp gravity waves in the top of the simulation domain in a so-called buffer layer. The tendency associated with the damping at grid cell $i, j, k$ is calculated for an arbitrary variable $\phi$ as

$$\frac{\partial \phi}{\partial t}\bigg|_{i,j,k,\text{buf}} = \frac{\phi_{i,j,k} - \phi_{0;k}}{\tau_{d;k}} \tag{94}$$

where $\phi_0$ is taken from a user-specified reference profile, and time scale $\tau_d$ is a measure for the strength of the damping. It varies with height and is calculated at height $z_k$ following

$$\tau_{d;k}^{-1} = \sigma\left(\frac{z_k - z_{b;bot}}{z_{b;top} - z_{b;bot}}\right)^\beta, \tag{95}$$

where $\sigma$ is the damping frequency chosen by the user and $\beta$ an exponent that describes the shape of the vertical profile of the damping frequency, which is always zero at the bottom ($z_{b;bot}$) and $\sigma$ at the top ($z_{b;top}$) of the layer.

## 5    Model output

### 5.1    Statistics

A large set of output statistics can be computed during runtime at user-specified time intervals. The statistics module provides vertical profiles of means, second-, third- and fourth-order moments of all prognostic variables, as well as advective and diffusive fluxes. Furthermore, there are multiple diagnostic variables, such as the pressure, the pressure variance and its vertical

flux. The thermodynamics generate their own statistics based on the chosen option. The moist thermodynamics provides a large set of cloud statistics.

There is a separate module for budget statistics, which provides the budgets of all components of the Reynolds stress tensor, and those of the variance and vertical flux of the thermodynamic variables.

One of the key features of the MicroHH statistics routine is that an arbitrary mask can be passed into the routine over which

the statistics are calculated. This allows, for instance, for a very simple way of computing conditional statistics in updrafts or clouds, which is demonstrated later in Section 9.2.



## 5.2 Two- and three-dimensional output

It is possible to save two-dimensional cross sections and three-dimensional fields of any of the prognostic and diagnostic variables of the model, as well as of derived variables. This output can be made at user-specified time intervals. Cross sections can be made in any chosen $xy-$, $xz-$, and $yz-$plane. Derived variables (any arbitrary function of existing model variables), can be easily added to the code by the user.

## 6 Technical details of the code

### 6.1 Code structure

MicroHH is written in C++ and uses elements of object-oriented programming and template metaprogramming. The code components are written in classes that define the interface. Inheritance is used to allow for specializations of classes. This way of organizing the code has two advantages: it minimizes switches and it allows code components and their extensions to reside in their own file, which increases code clarity and facilitates the merging of new code extensions. High performance of computational kernels is achieved by executing kernels in their own function, with explicit inclusions of the `restrict` keyword to notify the compiler that fields do not overlap in memory. Furthermore, compiler-specific pragmas are used to aid vectorization on Intel compilers.

### 6.2 GPU

MicroHH is enabled to run on fast and energy-efficient Graphical Processing Units (GPU). This promising technique has been pioneered in atmospheric flows by Schalkwijk et al. (2012) and has shown its potential in weather forecasting (Schalkwijk et al., 2015). The implementation is based on NVIDIA's CUDA. The CPU and GPU version reside in the same code base, where the GPU implementation is activated with the help of precompiler statements. The philosophy is that the entire model is initialized on the CPU and that the GPU implementation is only activated right before starting the main time loop. At that stage the required fields are copied in double precision accuracy to the GPU, and the entire time integration is done there. Synchronization only happens when statistics have to be computed or when restart files or cross sections of flow fields are saved to disk, to ensure high performance.

### 6.3 Parallelization

The code uses the Message Passing Interface (MPI) in order to run on a large number of cores. The three-dimensional simulation domain is split into vertically-oriented columns standing on a two-dimensional grid.

The code assigns one MPI task to each grid column using the MPI_Cart_create function, and uses this grid to detect the IDs of neighboring processes. In order to avoid complex packing routines, we make use of MPI datatypes wherever possible. The MPI calls are hidden in an interface to avoid the need to manually write MPI calls.





The input/output (IO) is entirely based on MPI-IO to ensure that three-dimensional fields and two-dimensional cross sections are stored as single files. We have chosen MPI-IO in order to limit the number of files resulting from simulations on a large number of processes and to allow for restarts on a different number of processes. In order to keep complexity of the IO as low as possible, we make use of the MPI_Sub_array function in combination with MPI_File_write_all in order to write the fields.

## 6.4  External dependencies

MicroHH depends on several external software tools or libraries. It uses the CMake build system for the generation of Makefiles. CMake allows for parallel builds, which minimizes the compilation time, and it is easy to add configurations for different machines. Furthermore, the FFTW3 library (Frigo and Johnson, 2005) is used for the computation of fast-Fourier transforms. The statistical routines make use of netCDF software developed by UCAR/Unidata[1]. In order to run the provided test cases and their output scripts, a Python installation including the NumPy and Matplotlib modules is required. Automatic documentation generation can be done using Doxygen, but this is optional.

## 7  Running simulations

In order to run a simulation with MicroHH, a sequence of steps needs to be taken. Each simulation has an `.ini` file that contains the settings of the simulation, a `.prof` file that contains the (initial) vertical profiles of all required variables, and, if time-varying boundary conditions are desired, a file with the prescribed time evolution for all time-varying boundary conditions. To prepare a simulation with the name `test_simulation`, MicroHH needs to be run in the following way

```
./microhh init test_simulation
```

where it is assumed that `test_simulation.ini` and `test_simulation.prof` are available in the directory where the command is triggered. This procedure will create the initial fields of all prognostic variables and save the required fields for those model components that need to save their state to guarantee bitwise identical restarts.

After the previously described initialization phase, the execution of

```
./microhh run test_simulation
```

will start the actual simulation. Depending on the chosen output intervals, the simulation will create restart files, statistics, cross sections, and field dumps. MicroHH can restart from any time at which the restart files are saved.

The last mode in which the code can run is the post-processing mode. By running

```
./microhh post test_simulation
```

the code will generate statistics from saved restart files at a specified time interval. This mode allows the user to create new statistics and calculate those from saved data, without having to rerun the simulation.

---

[1]http://doi.org/10.5065/D6H70CW6





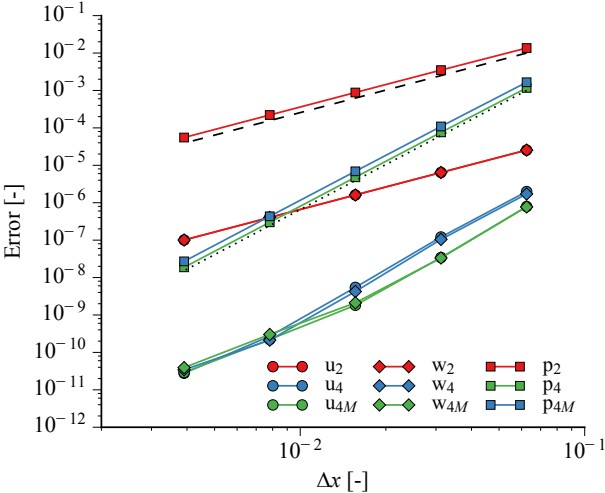

**Figure 1.** Convergence of the spatial discretization error in the two-dimensional Taylor-Green vortex. The dashed black line is the reference for second-order convergence, the dotted black line for fourth-order convergence.

## 8 Validation of the dynamical core

In this section, we present a series of cases intended to validate MicroHH under a wide range of settings. Each of these test cases is available in the `cases/` directory in the MicroHH repository, where all detailed settings can be found (see Sect. 13). Below, we present only the most relevant information per case.

### 8.1 Taylor-Green vortex

The two-dimensional Taylor-Green vortex (`cases/taylorgreen`) presents an ideal test case for a dynamical core as it has an analytical solution, even though it is nonlinear. This flow is composed of two rotating vortices whose evolution in a domain $[0, 1; 0, 0.5]$ is described with

$$u(x,z,t) = \sin(2\pi x)\cos(\pi z)f(t), \tag{96}$$

$$w(x,z,t) = \cos(2\pi x)\sin(\pi z)f(t), \tag{97}$$

$$p(x,z,t) = \tfrac{1}{4}\left(\sin(4\pi x) + \sin(4\pi y)\right)f(t)^2, \tag{98}$$

where $f(t) = 8\pi^2\nu t$.

We use the analytical form at $t = 0$ as the initial condition and run this case for one vortex rotation ($t = 1$), with $\nu = (800\pi^2)^{-1}$. We compare the result against the analytical solution for a set of grid spacings and with the second-order and fourth-order dynamical core; for the latter we compare the most accurate advection scheme and the fully energy-conserving one.

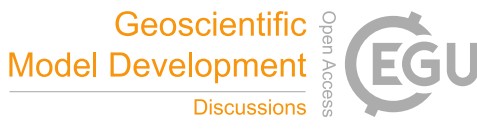



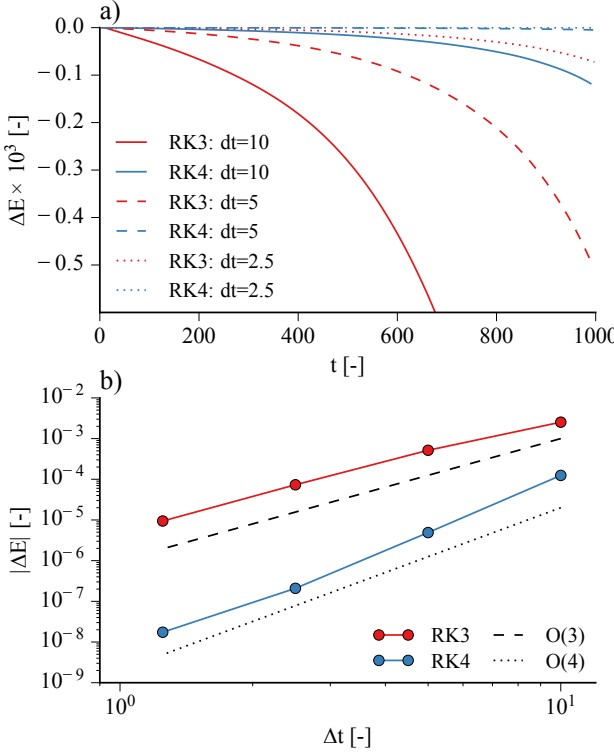

**Figure 2.** Time evolution of the energy loss during 1000 time units of random noise advection for the RK3 and RK4 time integration schemes with three different time steps (a). Error convergence of the temporal discretization for the RK3 and RK4 scheme (b).

Figure 1 shows the error convergence of the simulations. The error for a variable $\phi$ is computed as $\sum \Delta x \Delta z \left| \phi_{i,k} - \phi_{\mathrm{ref},i,k,} \right|$ over the two-dimensional domain, where $\Delta x$ and $\Delta z$ are the uniform grid spacings used in this case and $\phi_{\mathrm{ref}}$ is the analytical solution. All variables converge according to the order of the numerical scheme. The fourth-order dynamical core loses accuracy at fine grid spacings. This is due to the boundary condition for the vertical velocity that sacrifies an order of accuracy to

5    ensure global mass conservation (Morinishi et al., 1998).

### 8.2   Energy conservation and time accuracy

The second test of the dynamical core consists of combined evaluation of energy conservation and time accuracy (`cases/conservation`). In this experiment, we switch the diffusion off and advect random noise through the domain for 1000 seconds. These tests have been conducted with the third- and fourth-order Runge-Kutta schemes. We have chosen for the

10    fourth-order spatial discretization in order to demonstrate its energy conservation.

The loss of energy as a function of time is shown in Fig. 2a. The fourth-order scheme results in a smaller energy loss for the same time step and an a faster convergence. The error-convergence plot (Fig. 2b) shows that the error convergence is in





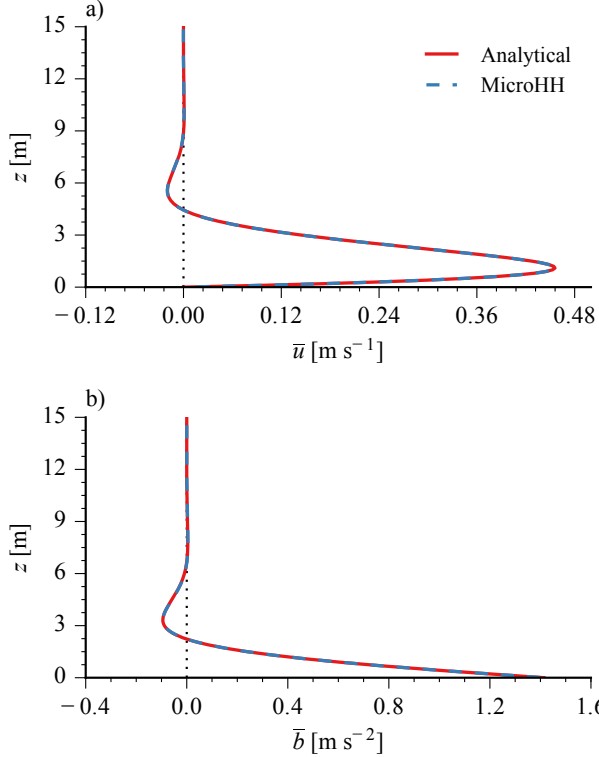

**Figure 3.** Normalized numerical Prandtl-model solutions for velocity $u$ (left) and buoyancy $b$ (right) compared to their analytical counterparts.

accordance with the order of the respective scheme. Furthermore, it illustrates the fact that, if high accuracy in time is desired, the five-stage fourth-order scheme is less expensive than the three-stage third-order scheme. For instance, at a $\Delta t$ of 10, the error of the fourth-order scheme is approximately equal to the error of the third-order scheme at a $\Delta t$ of 2.5. To reach this accuracy, the fourth-order scheme needs only 5 steps per 10 time units, whereas the third-order scheme needs 12 steps.

5  ### 8.3   Laminar katabatic flow

To test the buoyancy routine and the option to put the domain on a slope, a laminar katabatic flow has been simulated, based on the test case of Prandtl (1942) (`cases/prandtl`). In this test case, the surface is inclined at an angle of $30°$ and a linearly stratified atmosphere ($N = 1\,\mathrm{s}^{-1}$) is cooled from below with a fixed surface buoyancy flux of -0.005 m$^2$ s$^{-3}$.

The fluid, which was initially at rest, goes through a series of decaying oscillations after the negative buoyancy flux is applied
10  at the surface. Eventually, it reaches the steady state corresponding to the Prandtl model solution. Numerical integration was performed sufficiently long for the oscillation amplitude to become a small fraction of the amplitude of the first oscillation. Comparison of analytical and numerical solutions, which very closely agree with each other, is presented in Fig. 3.





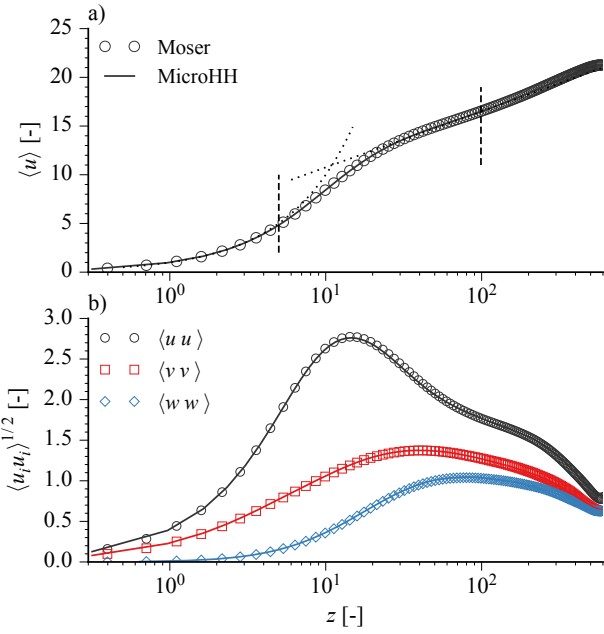

**Figure 4.** Velocity means and variances for Moser et al. (1999) channel flow case at Reynolds-$\tau$ 590. The dashed vertical lines marks the spectra locations. Height $z$ is normalized with $u_\tau/\nu$, velocities with $u_\tau^{-1}$.

## 8.4 Turbulent channel flow

For fully turbulent flows, the numerical solutions cannot be compared against any analytical testcases. Therefore, we validate our results against a channel flow at a Reynolds-$\tau$ number of 590 (Moser et al., 1999) for means, variances, spectra, and second-order budget statistics (`cases/moser590`). The case is run at a resolution of $768 \times 384 \times 256$ grid points.

Figure 4a shows the normalized horizontally-averaged streamwise velocity. The normalized rms of all three velocity components are presented in Fig. 4b. All plotted variables show a perfect match with the data and are indistinguishable from Moser's data. In order to further assess the accuracy of the data, we show the second-order budgets of the variances in Fig. 5. Also here, the match with the reference data is excellent, which indicates that the whole range of spatial scales in the flow is represented well and that the fourth-order scheme is well able to pick up the small scale details of the flow.

The findings in the previous paragraph are further corroborated by the spectra shown in Fig. 6. Over the whole range of scales, the match between our simulation and that of Moser is excellent.

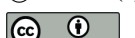



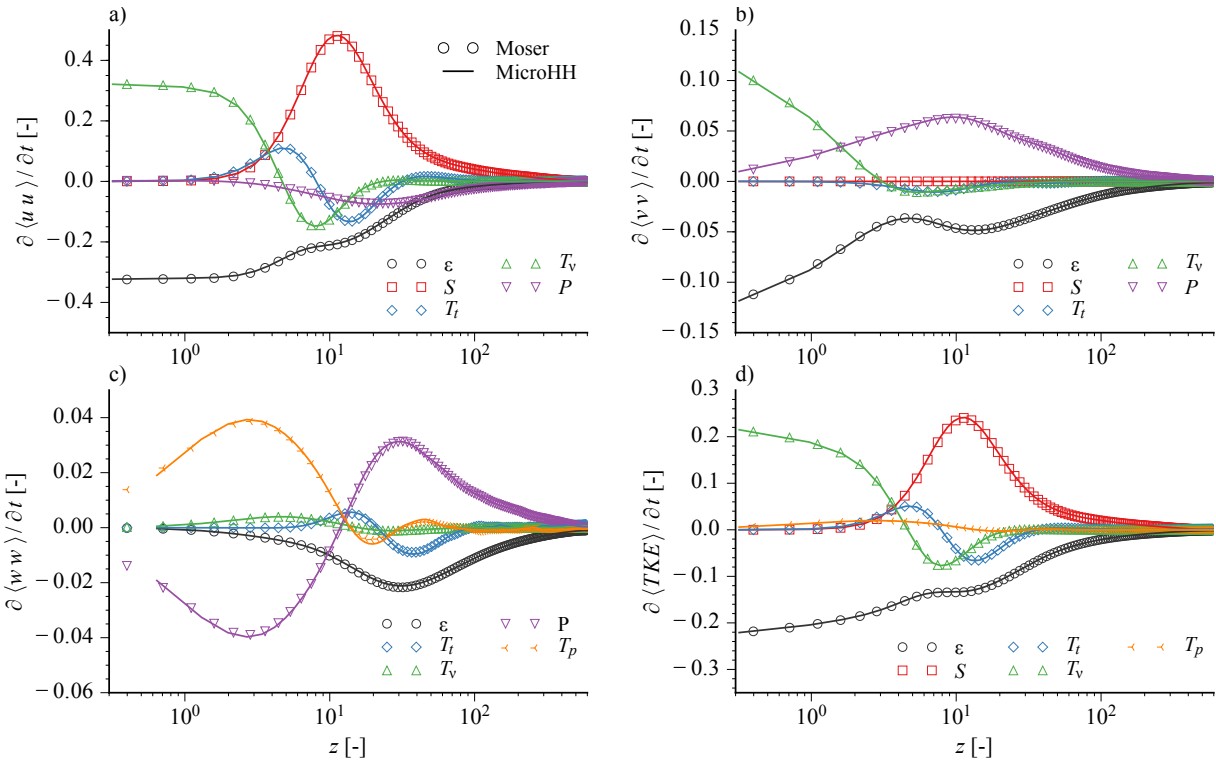

**Figure 5.** Budgets of variances and TKE compared against Moser et al. (1999)'s reference data at Reynolds-$\tau$ of 590. Height $z$ is normalized with $u_\tau/\nu$, the variances and TKE budget with $\nu/u_\tau^4$.

## 8.5 Turbulent katabatic flow

The final evaluation of the dynamical core without parametrizations enabled is based on the direct numerical simulation of a turbulent katabatic flow. Here, a buoyancy driven slope flow is simulated following the setup of Fedorovich and Shapiro (2009) (`cases/drycblslope`).

5     A flow over a slope inclined at an angle $\alpha$ of 60° is simulated with a fixed uniform surface buoyancy flux of -0.5 m² s⁻³. The simulation is performed in a domain of 0.64 m × 0.64 m × 1.6 m using a uniform grid of 256 × 256 × 640 points. The initial state is a fluid at rest with a linear buoyancy stratification $N$ of 1 s⁻¹. No-slip boundary conditions are applied at the bottom, free-slip at the top.

    Turbulent motion starts quickly after the buoyancy flux is applied at the surface. It is characterized by random, large-
10 amplitude fluctuations of velocity and buoyancy fields in the near-slope core region, and shows a quasi-periodic oscillatory behavior at larger distances from the slope. Mean profiles of along-slope velocity component and buoyancy, as well as profiles of second-order turbulence statistics, such as kinematic turbulent fluxes of momentum and buoyancy, and velocity-component





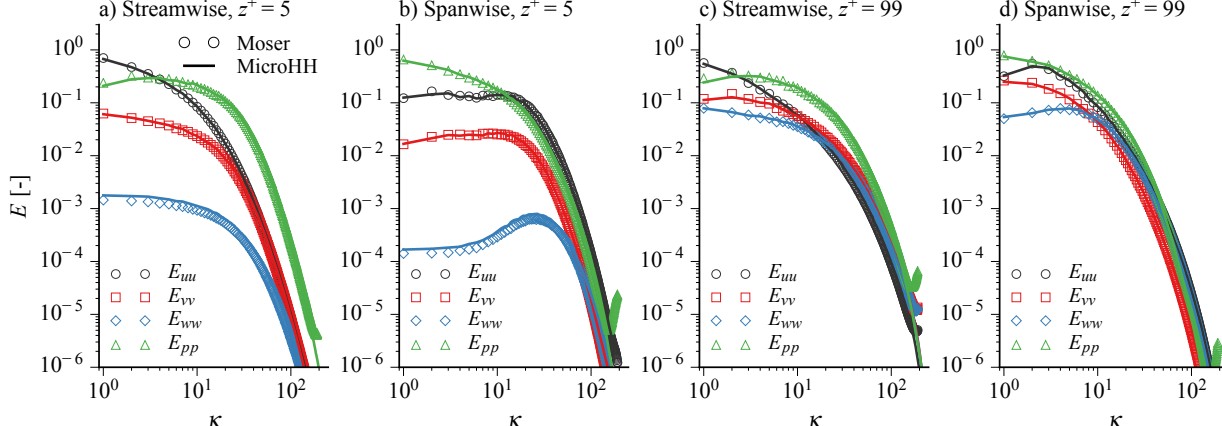

**Figure 6.** Spectra of all velocity components and pressure compared against Moser et al. (1999)'s reference data at Reynolds-$\tau$ of 590. The velocity spectra are normalized with $u_\tau^{-2}$, the pressure spectra with $u_\tau^{-4}$

and buoyancy fluctuation variances, were evaluated by averaging the simulated flow fields spatially over the along-slope planes and temporally over five oscillation periods beyond the transition stage.

For comparison, the same katabatic flow case was reproduced using the numerical code (hereafter referred to as FS09) that was employed to simulate turbulent slope flows in Shapiro and Fedorovich (2008) and Fedorovich and Shapiro (2009). In that

code, the time advancement was performed with an Asselin-filtered second-order leapfrog scheme (Durran, 2013). The spatial discretizations are identical to the second-order accurate ones of MicroHH.

Numerical results obtained with both numerical codes testify that stable environmental stratification in combination with negative surface buoyancy forcing in the katabatic flow leads to an effective suppression of vertical turbulent exchange in the flow region adjacent to the slope. This suppression results in a shallow near-surface sublayer with strong buoyancy gradients

(Fig. 7a) capped by a narrow jet with peak velocity located very close to the ground (Fig. 7b). Further comparison has been performed on the slope-normal fluxes of momenum and buoyancy (not shown), where a nearly perfect match has been reproduced as well.

## 9   Validation of atmospheric large-eddy simulations

### 9.1   Dry convective boundary layer with strong inversion

As a first test case of MicroHH in LES mode, we present that of Sullivan and Patton (2011) (`cases/sullivan2011`). This is a dry clear convective boundary layer that grows into a linearly stratified atmosphere with a very strong capping inversion (see Fig. 8a). The system is heated from the bottom by applying a constant kinematic potential temperature flux of 0.24 K m s$^{-1}$. The domain size is 5120 m $\times$ 5120 m $\times$ 2048 m. Gravity wave damping has been applied in the top 25 percent of the domain.





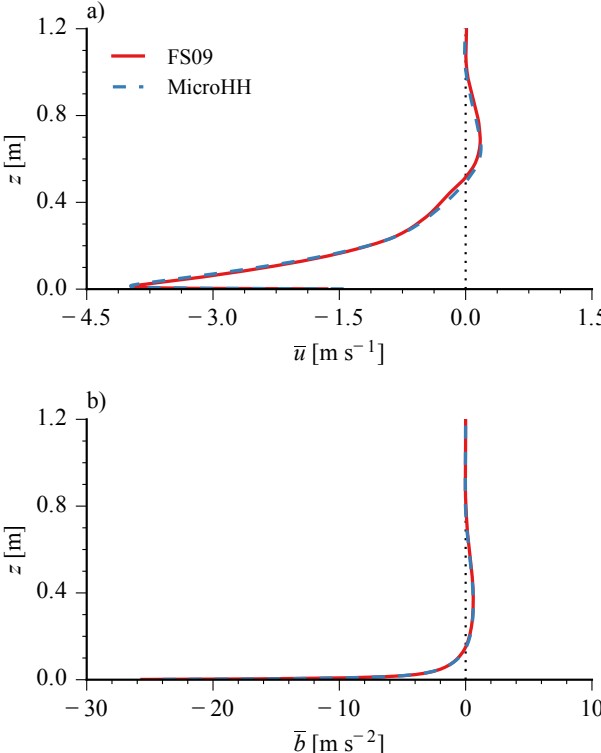

**Figure 7.** Profile of the mean along-slope velocity (a) and buoyancy (b) as predicted by MicroHH and FS09.

Simulations have been run for three hours with three spatial resolutions. The time-averaged profiles have been calculated over the last hour.

The results show the formation of a well-mixed layer with an overlying capping inversion (see Fig. 8a) and the associated linear heat-flux profile with negative flux values in the entrainment zone (see Fig. 8b). The change of both quantities with resolution highlights the intrinsic challenge in resolving a boundary layer with an inversion layer that is stronger than the numerical schemes can accurately resolve. At coarse resolution, the strong inversion leads to an unphysical overshoot of the potential temperature flux above the boundary layer top (see Fig. 8b). This overshoot leads to a numerical mixed layer on top of the entrainment zone that disappears quickly with increasing resolution.

### 9.2 BOMEX

The BOMEX shallow cumulus case (Siebesma et al., 2003) (`cases/bomex`), S03 hereafter, provides the opportunity to evaluate the moist thermodynamics (see Sect. 3.8) and large-scale forcings. We have repeated the case as described in S03 at the original resolution of the study (100 m × 100 m × 40 m) and at a higher resolution (10 m × 10 m × 9.375 m).



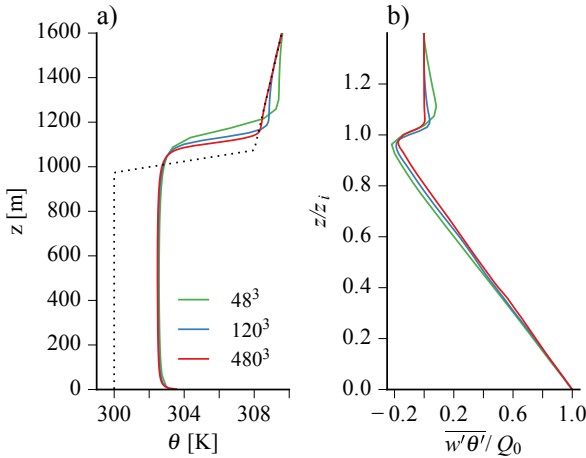

**Figure 8.** Vertical profiles of horizontally-averaged potential temperature (a) and normalized kinematic heat flux (b). The boundary layer depth $z_i$ is the location of the maximum vertical gradient in the potential temperature profile shown in (a).

This case produces non-precipitating shallow cumulus. It has a large-scale cooling applied that represents radiation, as well as a large-scale drying to allow the atmosphere to relax to a steady state. In addition, a large-scale vertical velocity is applied over a certain height range to reproduce the appropriate synoptic conditions.

The simulation is run for 6 hours. Statistics are recorded during the final hour, including conditional statistics for the cloud-covered fields ($q_l > 0$) and for the cloud cores ($q_l > 0$ and $\theta_v > 0$). The vertical profile of area coverage and profiles of horizontally-average liquid water potential temperature $\theta_l$, total water $q_t$, and vertical velocity $w$ are shown in Fig. 9. All mean and conditionally sampled statistics at the original resolution are predominantly within one standard deviation of the ensemble mean of data from all models that participated in S03. The horizontally-averaged vertical velocities in the cloud and cloud core decrease considerably with an increase in resolution.

## 9.3 GABLS1

To evaluate the LES mode for stable atmospheric conditions, the GABLS1 LES intercomparison case (Beare et al., 2006) (`cases/gabls1`) was reproduced. The boundary layer develops in this case from a shallow well-mixed layer into a weakly stable boundary layer, driven by a prescribed negative tendency of the surface temperature over a total integration time of 9 hours. The Boussinesq approximation is used, the advection scheme uses fourth-order accurate interpolations (Eq. 27), and the Smagorinsky subgrid turbulence scheme is set up with a Smagorinsky constant equal to 0.12, and a subgrid turbulent Prandtl number of unity. The experiments are performed at two different resolutions with grid cells of 2 m and 6.25 m, and compared to the models which participated in the study of Beare et al. (2006).



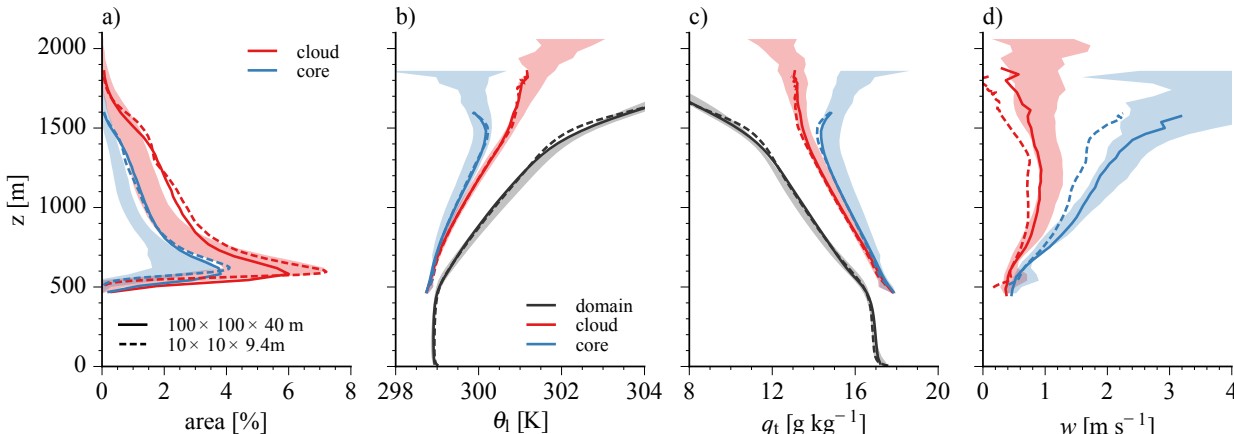

**Figure 9.** BOMEX LES intercomparison (S03). Shown are the domain mean, and conditionally sampled cloud ($q_l > 0$) and cloud core ($q_l > 0$ and $b - \langle b \rangle > 0$) vertical profiles of (a) area coverage, (b) liquid water potential temperature, (c) total specific humidity and (d) vertical velocity. The results are averaged over $t = 18000$ s – 21600 s. The shaded area denotes the mean $\pm$ one standard deviation of the participating models from S03, the solid and dashed lines the results from MicroHH, using the original (solid) and a higher resolution (dashed) setup.

Figure 10 shows the domain and time-averaged (over a period from 28800 to 32400 s) vertical profiles of potential temperature ($\langle \theta \rangle$) and the velocity component ($\langle u \rangle$), and also time series of the boundary layer depth ($z_{ABL}$) and friction velocity ($u_*$). At the largest grid spacing of 6.25 m, it takes approximately 2 hours for the flow to become turbulent, as is evident from the delayed boundary layer growth and abrupt changes in $u_*$. Nonetheless, typical features like the low-level jet (Fig. 10b) are

5  well reproduced, and all statistics are predominantly within the range of results from Beare et al. (2006). With the grid spacing reduced to 2 m, the flow becomes turbulent nearly instantaneously, but the resulting boundary layer depth and surface friction velocity are on the low side compared to the 5 models from Beare et al. (2006) which were run at this resolution.

# 10  Performance

## 10.1  CPU

10  The parallel performance of MicroHH has been evaluated in strong- (`cases/strongscaling`) and weak-scaling (`cases/weakscaling`) experiments. The case used is direct numerical simulation of a buoyancy driven atmospheric boundary layer based on van Heerwaarden et al. (2014). For each simulation in the scaling experiments, a series of time steps is performed, and the mean cost per step is calculated over the series. The strong-scaling experiment has been performed on LRZ's SuperMUC[2] machine. In this experiment simulations were performed on $1024 \times 1024 \times 1024$ and $2048 \times 2048 \times 1024$

---

[2]https://www.lrz.de/services/compute/supermuc/

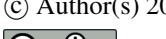



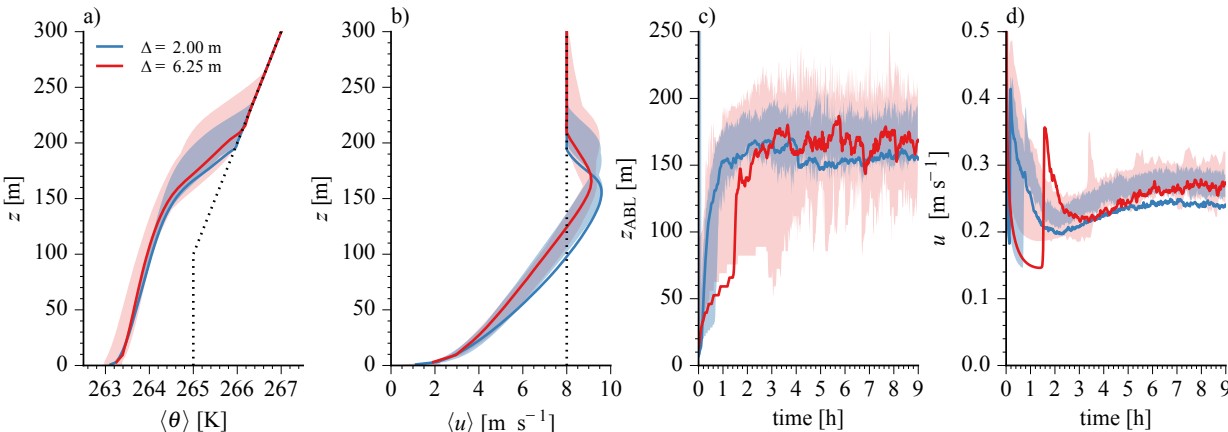

**Figure 10.** GABLS1 LES intercomparison (Beare et al., 2006). Shown are the vertical profiles of (a) potential temperature and (b) $u$-component of the velocity, and time series of the (c) boundary layer depth and (d) surface friction velocity. The shaded areas show the range in the results from the models that participated in the Beare et al. (2006) study. The dotted black lines show the initial conditions.

grid points, with the number of processes increased throughout the scaling experiment. The weak-scaling experiment has been performed on Forschungszentrum Jülich's Juqueen[3] machine. In this experiment, a fixed $64 \times 32 \times 1024$ grid is assigned to each processor and throughout the experiment the domain size is increased. The results of both experiments are shown in Figure 11.

The strong-scaling experiment shows that increasing the number of processors leads to faster simulations. The speedup is initially close to linear, but each consecutive increase in the number of cores makes the model less efficient. Based on these results, we conclude that for the chosen test case and for the used supercomputers, a work load of approximately $2 \times 10^6$ grid points per core is the best balance between speed and computational efficiency.

The weak scaling shows almost 90 percent efficiency going from 512 to 8192 cores, beyond that the scaling falls off to 80 percent. This can be explained by physical properties of the machine; beyond 8192 cores a simulation no longer fits on one midplane (a physical unit consisting of 8192 cores), leading to slower communication.

## 10.2 Performance GPU (CUDA) implementation

The GPU implementation of MicroHH allows for fast simulations on a single device. The current state-of-the-art GPUs feature 12 GB of memory, thus simulations of maximally $512 \times 512 \times 512$ grid points of a flow with three velocity components, pressure, two scratch fields for temporary storage, and a single scalar fit in memory. To test the performance of such simulations, the performance of MicroHH on an NVIDIA Quadro K6000 (using CUDA 6.5) has been compared against the Max Planck Institute for Meteorology's cluster Thunder (2 Intel Xeon E5-2670 CPU's per node, 16 cores per node, Intel 15.01 with OpenMPI 1.8.4).

---

[3]http://www.fz-juelich.de/ias/jsc/EN/Expertise/Supercomputers/JUQUEEN/JUQUEEN_node.html





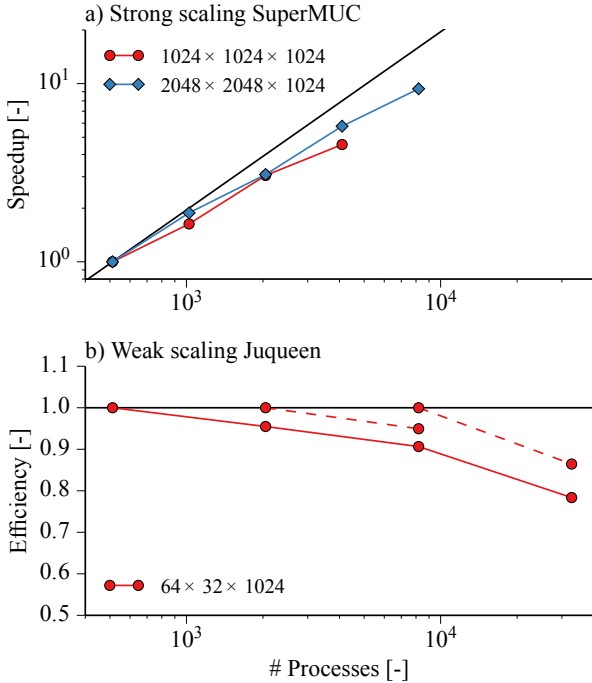

**Figure 11.** Speed-up from a strong-scaling experiment (a). Efficiency from a weak-scaling experiments (b). Black lines indicate perfect speed-up and efficiency. The dashed red lines show the efficiency change relative to the previous measurement.

**Table 1.** Speedup of GPU simulation compared to respective CPU simulation performed on $n$ cores.

| case | $n=1$ | $n=16$ | $n=32$ | $n=64$ |
|------|-------|--------|--------|--------|
| B64  | 18.49 | 1.93   | 1.14   | 0.95   |
| B128 | 28.01 | 2.98   | 1.51   | 0.92   |
| B256 | 27.76 | 3.02   | 1.59   | 0.91   |
| B512 | 29.88 | 3.03   | 1.56   | 0.86   |
| M180 | 21.57 | 2.17   | 1.13   | 0.69   |
| M600 | 22.55 | 2.25   | 1.06   | 0.60   |

Three benchmark cases have been chosen: the BOMEX moist convection case on grids of $64^3$, $128^3$, $256^3$ and $512^2 \times 384$, and the channel flow cases of Moser et al. (1999) at a Reynolds-$\tau$ number of 180 and 590.

The results shown in Table 1 point to the great potential of GPU computing. For the considered cases, which all fit on a single GPU, it takes at least 32 cores to reach equal performance. Only at 64 cores, the CPU simulations are notably faster.



Therefore, for simulations that fit into its memory, the GPU provides an excellent alternative for the CPU, especially as the GPU is very energy efficient.

## 11  Future plans

There are several ongoing projects to extend the model. Currently, a parameterizations for microphysics has been developed, and an interactive land surface model is under development. In addition, the immersed boundary method following Tseng and Ferziger (2003) is being implemented to allow for simulations of flow over obstacles and hills.

Furthermore, preliminary experiments have been performed to include a Domain-Specific Language (DSL) to enable the expression of complex finite difference operators in simple and compact syntax (https://github.com/Chiil/stencilbuilder/). This development has shown great potential, for two reasons. First, the DSL prevents implementation errors, as the explicit indexing in computational kernels with spatial operators can be omitted. Second, the DSL allows for simple implementation of system-specific tuning, such as loop tiling or OpenMP.

## 12  Conclusions

This paper has presented a full description of MicroHH, a new computational fluid dynamics code for simulations of turbulent flows in the atmospheric boundary layer. The governing equations and their implementation has been presented, and a broad validation under a wide range of settings has been shown.

## 13  Availability of code and resources

MicroHH has its own website at http://microhh.org. The code is hosted at GitHub and can be accessed either via the website, or directly from https://github.com/microhh/microhh. The GitHub website includes a wiki with several tutorials, including one to compile and run the code. The GitHub repository is coupled to Zenodo, which provides DOIs for released software. The release on which the reference paper is based is found at https://zenodo.org/badge/latestdoi/14754940. A selection of visualizations can be viewed at the MicroHH channel on Vimeo https://vimeo.com/channels/microhh/.

## Appendix A: Appendix

Table 2 presents an overview of the chosen values for physical constants in the code.

*Acknowledgements.* Finishing the 1.0 version of MicroHH has only become possible thanks to extensive discussions and exchange of code with many of the developers of DALES (Huug Ouwersloot, Stephan de Roode, Arnold Moene and Jordi Vilà-Guerau de Arellano), PALM (Björn Maronga and Siegfried Raasch), UCLA-LES (Linda Schlemmer and Bjorn Stevens) and ICON (Leonidas Linardakis). We thank Alan Shapiro for the discussions on the bottom boundary condition for pressure for buoyancy driven flows. The authors gratefully acknowledge the



**Table 2.** Overview of used constants.

| Symbol | Description | Value | Units |
|---|---|---|---|
| $\kappa$ | Von Karman constant | 0.4 | - |
| $g$ | Gravitational acceleration | 9.81 | m s$^{-2}$ |
| $c_{\mathrm{p}}$ | Specific heat of dry air at constant pressure | 1005 | J kg$^{-1}$ K$^{-1}$ |
| $p_{00}$ | Reference pressure | $1 \cdot 10^5$ | Pa |
| $R_{\mathrm{d}}$ | Gas constant for dry air | 287.04 | J K$^{-1}$ kg$^{-1}$ |
| $R_{\mathrm{v}}$ | Gas constant for water vapor | 461.5 | J K$^{-1}$ kg$^{-1}$ |
| $L_{\mathrm{v}}$ | Latent heat of vaporization | $2.5 \cdot 10^6$ | J kg$^{-1}$ |

Gauss Centre for Supercomputing (GCS) for providing computing time for a GCS Large-Scale Project on the GCS share of the supercomputer JUQUEEN at Jülich Supercomputing Centre (JSC).



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
