# Peer review of "MicroHH 1.0: a computational fluid dynamics code for direct numerical simulation and large-eddy simulation of atmospheric boundary layer flows"

_Geoscientific Model Development, 2017_

## Short Comment (SC1) · 26 Mar 2017

- there's a sign error in the definition of buoyancy b in the text right above equation (12). The buoyancy is proportional to minus rho-prime (missing a minus sign).

- the slope flow equations (14)-(16) are valid for an x-axis pointing upslope. If the x-axis points downslope you need to change the sign of the sine(alpha) terms. Might be good to either state this or (better yet), put in a small diagram showing the coordinate system with x pointing up the slope.

---

## Author Comment (AC1) · 30 Mar 2017

The manuscript has been replaced today with a version containing correct figure labels. Something apparently went wrong in the merging of our submitted file with the GMD logo and all axis labels went missing. The latest version has the correct labels, and has the correct sign of the buoyancy following the comment of Prof. Shapiro.
* * *

---

## Referee Comment (RC1) · E. Bou-Zeid (Referee) · 9 May 2017

The paper documents the development of a CFD code that can be used in DNS and LES mode and that is made available to the community. The authors have a broad range of expertise in the physical and numerical aspects of such codes and this new code is very well designed. It will provide a great tool for researchers and I expect it to be widely used.

Some comments:

1. Eq 11: It would be useful to explain what Q represents physically (phase change,

radiative divergence, . . .). Also it should be included in 13 since the authors also use it to represent sources of heat unrelated to evaporation/condensation.

2. Page 5, first few paragraphs of the section "Gird" and many other places in the text. The authors use too many paragraphs. Some should be consolidated. E.g. the first 2 paragraphs of this section should be joined.

3. Eq 28: So I presume here the authors use j as the vertical index. That should be specified. Also maybe at some point the authors should point out that only the bottom and top boundary conditions (is it detailed sufficiently?) need a special treatment like this since the other are periodic.

4. Eq 28 again: At some point later in the paper I thought the authors mention that with 4th order accurate scheme 2 ghost cells are needed. If that is so, why is there a need for a biased formulation in 28 that would only use one ghost cell below the surface.

5. Eq 36 and other places: it would be useful if for each of these options (2nd versus 4th order for example), the flag that controls it in the code input file is listed. This will make it easy for the user to see how to control these options.

6. Eq 41: tilde is later used for filtering. Maybe denote the intermediate velocity with something else like an asterisk.

7. The fact that the code is mainly periodic in the horizontal direction should be underlined earlier in the paper than it is now. Maybe in the abstract.

8. After Eq 47: please provide a reference to the "Thomas algorithm"

9. LES equations 63 and so on are only for very high RE, i.e. wall modeled LES. Please specify that. Also it would be simple to use the code as a finite Re LES code by keeping the viscous term in 63. Why is this not pursued?

10. "Surface Model" section. The authors only provide the LES surface model. This should be specified. Also better is to add a description of how the DNS wall boundary

condition is treated, presumably through a viscous wall stress. Also the language seems to suggest that the LES is only over rough walls. There is nothing that prevent the code from simulating a smooth surface using the z0 ($\sim \nu/u^*$) of a smooth wall. This should be clarified.

11. First line after eq 73: please add "kinematic" to the description of B0.

12. Eq 74: the application of a log law to each velocity component separately is an approximation so the equals here should be replaced by $\approx$. Also this is a LOCAL MOST wall model. This is not a trivial detail and should be specified and discussed briefly with references to papers that discuss the implications in more detail.

13. Eqs 87 and 88: why not use an explicit approach using the fluxes at the previous time step? This is commonly done and since the CFL condition is typically quite < 1 this should be ok? What are the advantages of an explicit approach?

14. Eq 90 is confusing. For example under steady state this almost looks like the pressure gradient is 0. Should the mean RHS <f1> be added? The fact that the pressure gradient force must balance the surface stress force under steady state should be stated.

15. Eq 93: is the momentum balance changed when a subsidence velocity is added to scalars?

16. Page 18 lines 9-11: please provide reference or URLs for these libraries and codes.

17. Figure 1: which of the blue or green is the energy conserving 4th order or the most accurate. Also did the authors describe the 2 methods using these names in the numerics section?

18. ALL figures look like they have problems with some axis labels (some minus signs appear) and so on, please improve quality. If all looks good on the authors computers check that the PDF appears the same on other machines.

[Figure]

19. Why include RK3 in the code release at all given the results?

20. Page 20 line 9, delete "for"

21. Figure 2: slope at smallest dt looks the same for RK3 and RK4, no?

22. Figure 4: symbols not appearing in legend.

23. Section 8.4: give some info about MOSER code for comparison.

24. Page 22 line 6-10: use of word "data" to describe MOSER results is not a good choice here.

25. Figure 6: clearly the spectra of MOSER have some noise or aliasing issues that should be mentioned.

26. Page 24 Line 17: here the authors use the term "potential temperature flux" but previous they used "buoyancy flux". Pick one since they mean the same thing in dry cases. I would suggest potential T flux since it is a more accurate physics description.

27. Figure 7: maybe use log scale for y.

28. Page 25 line 8: delete "quickly"

29. Figure 9a: area coverage of what? Updrafts? Please clarify.

30. Section 9.3 and in general how is the code initialized? Random perturbations are added to mean profiles? Did the author try alternative approaches to seed turbulence?

31. Section 10: please provide info about the machines in section 10.1 (interconnect speed, processors per node, memory per nodes, ...). These details are needed to understand code scaling.

32. Figure 11: x axis label should be "processors"

---

## Referee Comment (RC2) · Anonymous Referee #2 · 23 May 2017

(Based on gmd-2017-41-manuscript-version2.pdf)

The manuscript describes an anelastic atmospheric finite difference model, called MicroHH 1.0, with extensions to include buoyancy-driven turbulence. The code is open source, built on a C++ library, and uses MPI and CUDA. Some validation results are provided for doubly-periodic domains, with and without the moist turbulence model. Numerical convergence demonstrates 2nd-4th-order accuracy for the dry model, and results match idealized convective turbulence tests. Overall the manuscript is clearly written, reasonably comprehensive, and establishes a capability that can be used for

understanding turbulent convection with the addition of more realistic physics parameterizations.

High-level suggested revisions: - In sections 1-2, switching between anelastic and Boussinesq should be made clearer, and with what approximations. In the rest of the paper, it should be clear what "mode" each test is run in. - Claims of conservation should state the caveat that the simplified equations are in flux-conservation form, but that they are not fully mass- or energy-conservative (for example, looking at total mass, $\rho_0 + \rho'$, in equation 10, is not conservative). p7 l18 how is it "fully energy conserving"? - A little more discussion of why this discretization was chosen, and what its benefits/limitations are. A little extra information would be a good way to flesh out the conclusion and provide more context for the reader.

Detailed minor revisions: - p1 abstract: "code reaches speedups of more than . . . conventional code" running on what processors? Generally best to express it as a % of peak FLOPS and specific to the two architectures you compare in results. - p1 "approach the synoptic scales" remove the? to clarify, maybe add LES resolution (< 1km?) at "scales of 1000km or more"?

- p2 l3, "order codes"? Older codes? - p2 last intro paragraph . . . it is worth mentioning Sec 5 (output), and 7 (instructions to reproduce), to encourage others to do the same, maybe mention w/ sec 13 or even move those sections to the end? - p2 l18 "constant with height z" maybe restate $\rho_0(z)$ only to support eq. (2)?

- p3 Derivation of eq(4) should be either referenced or add an extra step . . . eq (5) should come first, for example, to introduce the potential temperature EOS that's substituted into eq(4). - p3 l20 perturbational pressure form - not conservative / does not match eq (2)?

- p4 l16, introduced N without an equation/definition?

- p6-7 I appreciate the compactness of the notation and clarity in presenting it.

[Figure]

- p7 eq (40), why not use a similarly compact 4th-order 5-pt wide stencil, instead of the larger 7-pt wide one?

- p9 DFT solver eq (45) is not clear ... assuming periodic bc's or cosine transform for Neumann bc's on pressure? Is there a reference for this approach? - p9 "hat" DFT notation conflicts with "average" notation on p6. - p9 eq(46-47) could mention "corresponding to eq(39-40) respectively" around l17-18? - p9 l24, Ah! That's a big assumption, periodic lateral boundaries. Should be moved up and stated prominantly, along with motivation/limitations. Now I understand why p4 l15 "periodic with slopes" was introduced

- p11 l12-16, is the model-top pressure constant in time or modified every time step? what value is used?

- p12 l5, is filtering actually applied in your algorithm, and if so, at what resolution? Do you do anything to prevent discrete aliasing of unresolved wavelengths? - p12 tilde variables conflict with tilde "intermediate velocity" in eq (41) - p12 eq (67) $S_{ij}$ subscript? and what's the definition of $S^2$? - p12 l21, $N^2$ definition here different than above p4 l16?

- p13-15 sec 4.2 ... is this a new atm turbulence model? The reference Wyngaard (2010) is an entire book, and it is not clear which tests warrant which boundary conditions, etc. p15 l5 is particularly confusing ... might be worth describing Obukhov length and its use as a stability/mixing parameter, and why a look-up table is needed.

- p15 l11, why would you not just include a background $U_f$ and define a perturbational velocity from that? That would be compatible with periodic bc's, guarantee mass conservation, etc. - p15 bottom "adveciton" should be "advection"?

- p17 l19. "precompiler statements"? Meaning #define of GPU CUDA code? Any thoughts or statements on maintaining the different code bases in your C++ framework?

- p18 top, MPI-IO should have a reference? - p18 l9, change netCDF footnote to reference? - p18, maybe sections 5-7 should be moved/merged with 13 or all in an appendix? - p18 l25, love the post-processing mode based on restart files!

- p19 eq (98) should "4 \pi y" be z? - p19 figure 1 / p20 l1 discussion . . . L1 error in 2D should asymptote to hˆ4, even with 3rd-order boundary errors (O(N) pts * O(hˆ3) boundary error vs. O(Nˆ2) pts O(hˆ4) interior error). Please explain? Also adding a 2nd set of dotted lines for 3rd- and 4th-order on the bottom set if u,v 4M fields will better show the break.

- p20 l8, "diffusion off" you mean viscosity, no source terms, etc. so that total energy should be conserved? What's your equation for "energy" in this test? - p20 l10, "its energy conservation." you mean improved? It doesn't conserve energy exactly. - p20 figure 2, maybe put top figure on log |\DeltaE| scale as well to distinguish the results better?

- p21, line 4. Isn't there a difference in maximum CFL for each as well? - p22, l6, "perfect match" . . . so perfect it's hard to see any difference at all. What do you attribute that too, since you have completely different discretizations, etc. How were the Moser 1999 results so similar? Could you quantify the differences, plot them, and explain them? - p24, l12 . . . ditto for "nearly perfect match" here. Fig 6 also shows a "kink" in E_pp at higher \kappa . Is it worth explaining?

- p26 l8, Fig 9a,d - why are the vertical velocities diverging with resolution?

- p29 bottom p30. By putting these on a single GPU, you are avoiding communication overheads for the GPU. Did you run 1 MPI rank on the GPU? Did you run "n" MPI ranks on the CPU? For the B512 run you are getting very good (90%?) strong scaling for 1-4 CPU nodes.

- p30 l4, "a parameterizations . . . has been" singular? - p30 section 12 . . . could add a more comprehensive summary, call out any limitations or tradeoffs.

---

## Author Response (AR1)

**Rebuttal to review of Dr. Bou-Zeid**

The authors thank the reviewer for his kind words in his opening paragraph. We will address his comments point-by-point.

- Eq 11: It would be useful to explain what Q represents physically (phase change, radiative divergence, . . .). Also it should be included in 13 since the authors also use it to represent sources of heat unrelated to evaporation/condensation.
   Q can be any source or sink of heat. Phase changes are excluded from Q, as the dry dynamics do not support those, and the moist dynamics are based on the liquid water potential temperature that is constant under phase changes. The reviewer is correct that sources and sinks need to be included in Eq. 13 as well, and we will do so in the revised manuscript.
- Page 5, first few paragraphs of the section "Gird" and many other places in the text. The authors use too many paragraphs. Some should be consolidated. E.g. the first 2 paragraphs of this section should be joined. We will carefully go through the text and merge paragraphs at the suggested location and wherever appropriate.
- Eq 28: So I presume here the authors use j as the vertical index. That should be specified. Also maybe at some point the authors should point out that only the bottom and top boundary conditions (is it detailed sufficiently?) need a special treatment like this since the other are periodic.
   The reviewer is correct that j is the vertical index. We will explicitly mention this in the revised manuscript. We will also include an explicit reference to the fact that only the vertical dimension needs a special treatment.
- 4. Eq 28 again: At some point later in the paper I thought the authors mention that with 4th order accurate scheme 2 ghost cells are needed. If that is so, why is there a need for a biased formulation in 28 that would only use one ghost cell below the surface. Many operations involve a sequential application of two operators. For instance, in the 4th-order diffusion, we compute the laplacian as the divergence of a gradient. In this operation, only the gradient can make use of both ghost cells, but the divergence cannot, and therefore relies on a biased operator at the wall.
- 5. Eq 36 and other places: it would be useful if for each of these options (2nd versus 4th order for example), the flag that controls it in the code input file is listed. This will make it easy for the user to see how to control these options. MicroHH comes with a document that lists all the available options. We have failed to mention this in the text and will add it to the revised manuscript. We will explain as well in the revised manuscript that the model defaults to the order of generated grid.
- 6. Eq 41: tilde is later used for filtering. Maybe denote the intermediate velocity with something else like an asterisk.

We will follow the suggestion of the reviewer to avoid confusion between filtered

variables and the intermediate velocity.

- 7. The fact that the code is mainly periodic in the horizontal direction should be underlined earlier in the paper than it is now. Maybe in the abstract.
  We agree with the reviewer that an earlier notification is necessary, because it clarifies both the grid description and the pressure solver. We will introduce it in the introduction of the revised manuscript.
- 8. After Eq 47: please provide a reference to the "Thomas algorithm" We will include a reference in the revised manuscript.
- 9. LES equations 63 and so on are only for very high RE, i.e. wall modeled LES. Please specify that. Also it would be simple to use the code as a finite Re LES code by keeping the viscous term in 63. Why is this not pursued? We will follow the reviewer's suggestion and mention that our LES is developed for very high Re. Extending our code to a finite Re LES code would be trivial, but has not been pursued yet. The reason is that most MicroHH users that run the model in LES-mode run atmospheric cases.
- 10. "Surface Model" section. The authors only provide the LES surface model. This should be specified. Also better is to add a description of how the DNS wall boundary condition is treated, presumably through a viscous wall stress. Also, the language seems to suggest that the LES is only over rough walls. There is nothing that prevent the code from simulating a smooth surface using the z0 (~  $v/u^*$ ) of a smooth wall. This should be clarified.

The description of the DNS boundary conditions is contained in 3.7, but we failed to make this clear to the reviewer. We will improve both Section 3.7 and 4.2 to clarify our implementation. The code could indeed specify the z0 of a smooth wall, but also here, it has not been implemented yet.

- 11. First line after eq 73: please add "kinematic" to the description of BO. Correct. We will add this.
- 12. Eq 74: the application of a log law to each velocity component separately is an approximation so the equals here should be replaced by ≈. Also this is a LOCAL MOST wall model. This is not a trivial detail and should be specified and discussed briefly with references to papers that discuss the implications in more detail. We will introduce the approximation symbol. Furthermore, we will discuss the results with respect to existing literature, such as the reviewer's paper in Physics of Fluids (2005).
- 13. Eqs 87 and 88: why not use an explicit approach using the fluxes at the previous time step? This is commonly done and since the CFL condition is typically quite < 1 this should be ok? What are the advantages of an explicit approach?</li>
  Using the fluxes of the previous time step is often a good solution, but can lead to inaccuracies under, for instance, free convection, where fluxes and wind speeds can change fast at the surface, or under conditions of changing stability. Our methods

have a 100% convergence guarantee under all conditions. Furthermore, it is based on a lookup table that starts searching from the value at the previous time step, which makes it a very fast procedure.

14. Eq 90 is confusing. For example, under steady state this almost looks like the pressure gradient is 0. Should the mean RHS <f1> be added? The fact that the pressure gradient force must balance the surface stress force under steady state should be stated.

With Eq. 90, we aimed to show that the forcing is just part of the total tendency (note the  $F_{p;ls}$  suffix). It is a definition rather than an equality. As we have failed to explain it properly, we will clarify this in the improved manuscript.

15. Eq 93: is the momentum balance changed when a subsidence velocity is added to scalars?

It is not. Solving the momentum balance in a doubly-periodic domain under subsidence conditions is a non-trivial exercise that deserves its own study. We follow the simplified treatment that is used in other codes such as DALES and UCLALES. We will add an additional explanation to the paper.

- 16. Page 18 lines 9-11: please provide reference or URLs for these libraries and codes. We will add URLs to the referenced libraries and tools.
- 17. Figure 1: which of the blue or green is the energy conserving 4th order or the most accurate. Also, did the authors describe the 2 methods using these names in the numerics section?

The green line is the energy-conserving discretization, whereas the blue line is the accurate one. Surprisingly, the energy-conserving discretization is in the Taylor-Green-vortex test case also the most accurate one, but this does not apply to all test cases. We forgot to explain the abbreviations in the legend of Figure 1, and will do so in the figure caption of the revised manuscript. Furthermore, we will improve the color scheme to ensure that all cases can be easily distinguished.

- 18. ALL figures look like they have problems with some axis labels (some minus signs appear) and so on, please improve quality. If all looks good on the authors computers check that the PDF appears the same on other machines. Something apparently went wrong in the process of adding the GMD logos to the manuscript. In the current online version, as well as in the revised manuscript, all labels are in order.
- 19. Why include RK3 in the code release at all given the results? We will keep the RK3 case for testing purposes and for potential extension with implicit-in-time diffusion in the future. The reviewer is correct that our tests show that the RK4 scheme is beneficial under all conditions.
- 20. Page 20 line 9, delete "for" We will fix the sentence.

- 21. Figure 2: slope at smallest dt looks the same for RK3 and RK4, no? It appears so. The lines are bumpy and the exact slopes are hard to extract. We hope nonetheless, that the reviewer is convinced about the difference in convergence and accuracy between the two methods.
- 22. Figure 4: symbols not appearing in legend.

We believe this is related to the previous problem (point 18) we had with the figure axes. In our current version, all symbols are visible.

- 23. Section 8.4: give some info about MOSER code for comparison. The code of MOSER is spectral with Chebychev polynomials in the non-periodic dimension. We will explain this in the revised manuscript.
- 24. Page 22 line 6-10: use of word "data" to describe MOSER results is not a good choice here.

We will refer to MOSER's result as "model output data", rather than "data" in the revised manuscript.

25. Figure 6: clearly the spectra of MOSER have some noise or aliasing issues that should be mentioned.

The spectra of MOSER display aliasing in the pressure data, most likely related to the velocity multiplications in the Poisson equation that solves for the pressure. We will make this clear in the text.

- 26. Page 24 Line 17: here the authors use the term "potential temperature flux" but previous they used "buoyancy flux". Pick one since they mean the same thing in dry cases. I would suggest potential T flux since it is a more accurate physics description. We distinguish between the two. The dry dynamics have potential temperature as the governing variable, therefore the bottom BC is a potential temperature flux. Our simplified thermodynamics use buoyancy as the governing variable, and therefore a kinematic buoyancy flux as the bottom BC. We will clarify the text.
- *27. Figure 7: maybe use log scale for y.* We will remake the figure with a log scale and introduce it into the revised paper.
- 28. Page 25 line 8: delete "quickly" We will remove the word "quickly".
- 29. Figure 9a: area coverage of what? Updrafts? Please clarify. We were referring to the area coverage of cloud and cloud-core that are contained in the legend. We will make this explicitly clear in the figure caption in the revised manuscript.
- 30. Section 9.3 and in general how is the code initialized? Random perturbations are added to mean profiles? Did the author try alternative approaches to seed turbulence?

The code is initialized with random perturbations over the mean profiles, which is

sufficient for convective cases. We have also the options of introducing large vortices that are more efficient in generating turbulence under neutral or stable conditions. We will explain these options in the revised manuscript.

31. Section 10: please provide info about the machines in section 10.1 (interconnect speed, processors per node, memory per nodes, ...). These details are needed to understand code scaling.

We will introduce references to the machine specifications and introduce a brief description of each of them in the revised manuscript.

*32. Figure 11: x axis label should be "processors"* We will fix this in the revised manuscript and use the word "cores".

**Rebuttal to anonymous reviewer**

The authors thank the reviewer for his thorough review. We will first address the reviewer's high-level comments, and thereafter the detailed comments point-by-point.

**High-level suggested revisions:**

 In sections 1-2, switching between anelastic and Boussinesq should be made clearer, and with what approximations. In the rest of the paper, it should be clear what "mode" each test is run in. - Claims of conservation should state the caveat that the simplified equations are in flux-conservation form, but that they are not fully massor energy-conservative (for example, looking at total mass, \rho\_0 + \rho', in equation 10, is not conservative). p7 l18 how is it "fully energy conserving"? - A little more discussion of why this discretization was chosen, and what its benefits/limitations are. A little extra information would be a good way to flesh out the conclusion and provide more context for the reader. We will improve the revised manuscript with respect to the differences between Boussinesq and anelastic in the model implementation. In short, the implementation of the governing equations is the same under both approximations, but under Boussinesq, the reference density and potential temperature are constant with height in the momentum and mass-conservation equations.

Based on the reviewer's comments, we have not made our claims of energy conservation sufficiently clear. In Bannon (1996)'s anelastic approximation, the governing equations are energy conserving, in the sense that there is a correct transfer between kinetic and potential energy. This, however, does not mean that the discrete implementation is energy conserving. Our spatial discretization that follows Morinishi et al. (1998), conserves mass, momentum, and kinetic energy,

which we demonstrate in the paper. We will make the distinction between energy conservation in the governing equations and in the implementation clear throughout the improved manuscript.

**Detailed minor revisions:**

- p1 abstract: "code reaches speedups of more than ... conventional code" running on what processors? Generally best to express it as a % of peak FLOPS and specific to the two architectures you compare in results. We will explicitly mention in the abstract that it concerns single-GPU simulations and move the detailed information to the section on the scaling.
- p1 "approach the synoptic scales" remove the? to clarify, maybe add LES resolution (< 1km?) at "scales of 1000km or more"? We will follow the reviewer's suggestion and add some explicit numbers to the statement.
- 4. p2 l3, "order codes"? Older codes?"Order codes" will be changed to "other codes".

- 5. p2 last intro paragraph . . . it is worth mentioning Sec 5 (output), and 7 (instructions to reproduce), to encourage others to do the same, maybe mention w/ sec 13 or even move those sections to the end?
  We agree with the reviewer that all sections need to be mentioned. We are not sure what the reviewer means by "encouraging others to do the same" Does this refer to reproducing our test cases?
- p2 l18 "constant with height z" maybe restate \rho\_0(z) only to support eq. (2)? We will write rho\_0(z) instead of rho\_0, to make clear that rho\_0 is a function of height.
- 7. p3 Derivation of eq(4) should be either referenced or add an extra step ... eq (5) should come first, for example, to introduce the potential temperature EOS that's sub- stituted into eq(4). p3 l20 perturbational pressure form not conservative / does not match eq (2)?
  The reviewer is correct. We shall swap the order of the equation of state and the

The reviewer is correct. We shall swap the order of the equation of state and the momentum equation and cite the paper of Bannon (1996) earlier.

- p4 l16, introduced N without an equation/definition?
   We shall introduce the definition of N2 = db\_0/dz in the text, and do so for all thermodynamic modes.
- 9. *p6-71 appreciate the compactness of the notation and clarity in presenting it.* We appreciate the kind words of the reviewer. It has been a challenge to find a suitable notation.
- 10. p7 eq (40), why not use a similarly compact 4th-order 5-pt wide stencil, instead of the larger 7-pt wide one?

The 7-pt stencil has the advantage that it is built out of the same building blocks as the other operators, and thus uses the same ghost cells.

- 11. p9 DFT solver eq (45) is not clear ... assuming periodic bc's or cosine transform for Neumann bc's on pressure? Is there a reference for this approach? The DFT operator is only performed in the periodic x and y directions. Based on comments of the first reviewer, we will introduce earlier in the paper that our code is periodic in the two horizontal dimensions.
- 12. p9 "hat" DFT notation conflicts with "average" notation on p6.We will introduce a different symbol for the Fourier transform in the revised manuscript.
- 13. p9 eq(46-47) could mention "corresponding to eq(39-40) respectively" around l17-18?

We agree with the reviewer's suggestion and will refer to those equations.

14. p9 l24, Ah! That's a big assumption, periodic lateral boundaries. Should be moved up and stated prominantly, along with motivation/limitations. Now I understand why p4

**115 "periodic with slopes" was introduced.**

Following both reviewers' comments, we will introduce in the introduction that our code is doubly periodic.

15. p11 l12-16, is the model-top pressure constant in time or modified every time step? what value is used?

The model top pressure is the final result of the described procedure and depends on the surface pressure and the chosen reference profiles of temperature and humidity. MicroHH has the option of a constant profile in time, as well as a reference profile that updates in time.

- 16. p12 I5, is filtering actually applied in your algorithm, and if so, at what resolution? Do you do anything to prevent discrete aliasing of unresolved wavelengths?
  We do not use explicit filtering, but rely on the grid scale as a filter, which is a common procedure with atmospheric LES. With our numerical schemes, aliasing errors are small. We will introduce a short discussion on this in the revised manuscript.
- 17. p12 tilde variables conflict with tilde "intermediate velocity" in eq (41) We will use a different symbol in the revised manuscript.
- 18. p12 eq (67) S\_{ij} subscript? and what's the definition of S^2?In the revised manuscript we will write the full expression in terms of S\_{ij}. The reviewer is correct that we forgot the subscripts.
- 19. p12 l21, N2 definition here different than above p4 l16? The reviewer is correct. We have failed to make clear that depending on the chosen thermodynamics, an appropriate definition of N2 is used. We will clarify this in the improved manuscript, as mentioned in our reply to point 8.
- 20. p13-15 sec 4.2 . . . is this a new atm turbulence model? The reference Wyngaard (2010) is an entire book, and it is not clear which tests warrant which boundary conditions, etc. p15 I5 is particularly confusing . . . might be worth describing Obukhov length and its use as a stability/mixing parameter, and why a look-up table is needed.

We will clarify the text. The lookup table is only there for performance reasons, as it outperforms a Newton-Raphson method.

21. p15 l11, why would you not just include a background U\_f and define a perturbational velocity from that? That would be compatible with periodic bc's, guarantee mass conservation, etc.

By doing so, the problem will remain. If the large-scale pressure force is applied to the perturbation velocities only, it is no longer ensured that the perturbations average to zero, without applying the presented correction.

- 22. p15 bottom "adveciton" should be "advection"? The reviewer is correct.
- 23. p17 l19. "precompiler statements"? Meaning #define of GPU CUDA code? Any thoughts or statements on maintaining the different code bases in your C++ framework?

The use of precompiler statements is unavoidable, as we do not want to force the non-GPU user to install CUDA and compile the GPU code as well. We have chosen for an implementation in which the GPU code based is minimized, in order to ensure maintainability. We will elaborate our description of the CUDA implementation.

- 24. p18 top, MPI-IO should have a reference? We will introduce a reference.
- 25. p18 l9, change netCDF footnote to reference? We will introduce a reference.
- 26. p18, maybe sections 5-7 should be moved/merged with 13 or all in an appendix? We disagree with the reviewer here. We consider the presented topics in sections 5-7 of high relevance for a model description paper. Section 13 is located at the end of the paper following the GMD guidelines.
- 27. *p18 l25, love the post-processing mode based on restart files!* We thank the reviewer for this compliment. We would be very happy if this convinces the reviewer to use our code.
- 28. p19 eq (98) should "4 \pi y" be z? The reviewer is correct!
- 29. p19 figure 1 / p20 l1 discussion ... L1 error in 2D should asymptote to h4, even with 3rd-order boundary errors (O(N) pts \* O(h3) boundary error vs. O(N2) pts O(h4) interior error). Please explain? Also adding a 2nd set of dotted lines for 3rd- and 4th-order on the bottom set if u, v 4M fields will better show the break. In the 4th-order scheme, the boundary condition for vertical velocity w is set for global mass conservation rather than for 4th-order accuracy. We will add a second set of dotted lines to help the reader observe the convergence of the schemes.
- 30. p20 18, "diffusion off" you mean viscosity, no source terms, etc. so that total energy should be conserved? What's your equation for "energy" in this test?
  In this case, energy is kinetic energy. The model is run without viscosity and source terms, but with pressure solver to satisfy the continuity equation. We will clarify this in the improved manuscript.
- 31. p20 l10, "its energy conservation." you mean improved? It doesn't conserve energy exactly.

The spatial discretization does conserve energy, but it is the time discretization that

does not. We shall clarify this in the text.

32. p20 figure 2, maybe put top figure on log |\DeltaE| scale as well to distinguish the results better?

We prefer to keep our axis in its current form to show that our schemes are losing energy and therefore cannot lead to a blowup of the numerical solution. This is not possible if we plot the absolute value on a log scale. We shall clarify this in the improved manuscript.

- *33. p21, line 4. Isn't there a difference in maximum CFL for each as well?* There is. In this experiment, however, we chose to compare the accuracy that can be achieved a fixed time step, as this allows us to estimate the convergence.
- 34. p22, l6, "perfect match"... so perfect it's hard to see any difference at all. What do you attribute that too, since you have completely different discretizations, etc. How were the Moser 1999 results so similar? Could you quantify the differences, plot them, and explain them?

Both codes have fully converged results and are therefore identical if sufficient samples are averaged. Direct numerical simulation has, unlike LES, an exact solution, which makes the solution independent of the numerical schemes at sufficient resolution.

- 35. p24, l12 . . . ditto for "nearly perfect match" here. Fig 6 also shows a "kink" in E\_pp at higher \kappa. Is it worth explaining?
  MOSER has spectral schemes, which introduce aliasing errors in the highest wave numbers. Even though aliasing errors are removed when the nonlinear operators are applied, the solver for the pressure introduces new ones.
- 36. p26 l8, Fig 9a,d why are the vertical velocities diverging with resolution?This is often observed in LES-simulations of cumulus-topped boundary layers.Individual plumes that have a radius of only a few grid cells tend to overestimate velocity.
- 37. p29 bottom p30. By putting these on a single GPU, you are avoiding communication overheads for the GPU. Did you run 1 MPI rank on the GPU? Did you run "n" MPI ranks on the CPU? For the B512 run you are getting very good (90%?) strong scaling for 1-4 CPU nodes.

In our view, GPUs mostly deliver a benefit if simulations can be run on a single GPU. Therefore, we have taken one GPU. Furthermore, at the moment, MicroHH is only supporting single GPU simulations. We agree with the reviewer that our comparison might be unfair in the sense that the GPU simulation does not need communication, whereas the CPU simulation does. We have, however, decided to focus on simulation of sizes that are common in atmospheric LES studies. We will improve the discussion in the paper to make this clear.

- 38. p30 l4, "a parameterizations . . . has been" singular? The reviewer is correct!
- 39. p30 section 12 . . . could add a more comprehensive summary, call out any limitations or tradeoffs.We will elaborate the conclusions and highlight MicroHH's most important features

and limitations in the revised manuscript.

**MicroHH 1.0: a computational fluid dynamics code for direct numerical simulation and large-eddy simulation of atmospheric boundary layer flows**

Chiel C. van Heerwaarden1,2, Bart J. H. van Stratum1,2, Thijs Heus3, Jeremy A. Gibbs4, Evgeni Fedorovich5, and Juan-Pedro Juan Pedro Mellado2 1Meteorology 
[revised manuscript text omitted]
_{\underline{i,j}i,\underline{j,k}} \approx \overline{\phi}^{2x}_{\underline{i,j}i,\underline{j,k}} \equiv \frac{\phi_{i-\frac{1}{2},j,k} + \phi_{i+\frac{1}{2},j,k}}{2},$$
 (23)

$$\phi_{\underline{i,j}\underline{i,j,k}} \approx \overline{\phi}^{2xL}_{\underline{i,j}\underline{i,j,k}} \equiv \frac{\phi_{i-\frac{3}{2},j,k} + \phi_{i+\frac{3}{2},j,k}}{2}, \qquad (24)$$

Interpolations are marked with a hatbar. The superscript indicates the spatial order (2), and the direction (x) and has an extra qualifier L when it is taken using the wide stencil. The subscript indicates the position on the grid (i, j).

The gradient operators, denoted with letter  $\delta$ , are defined in a similar way

$$\frac{\partial \phi}{\partial x}\Big|_{\substack{i,j\,i,j,k\\ \longrightarrow}} \approx \delta^{2x}(\phi)_{\underline{i,j\,i,j,k}} \equiv \frac{\phi_{i+\frac{1}{2},j,k} - \phi_{i-\frac{1}{2},j,k}}{x_{i+\frac{1}{2}} - x_{i-\frac{1}{2}}}$$
(25)

$$\frac{\partial \phi}{\partial x} \bigg|_{\substack{i,j_{i},j,k \\ \to \infty}} \approx \delta^{2xL}(\phi) \underbrace{i,j_{i},j,k}_{i,j,k} \equiv \frac{\phi_{i+\frac{3}{2},j,k} - \phi_{i-\frac{3}{2},j,k}}{x_{i+\frac{3}{2}} - x_{i-\frac{3}{2}}}$$
(26)

We use the Einstein summation in the operators. For instance, the divergence of vector  $\frac{u_i|_{i,j}}{u_i|_{i,j,k}}$  can be written as 5  $\frac{\delta^{2x_i}(u_i)_{i,j}}{\delta^{2x_i}(u_i)_{i,j,k}}$ .

The fourth-order operators, written down in the same notation, are defined as

$$\phi_{\underline{i,j}\,\underline{i,j,k}} \approx \overline{\phi}^{4x}_{\underline{i,j}\,\underline{i,j,k}} \equiv \frac{-\phi_{i-\frac{3}{2},j,k} + 9\phi_{i-\frac{1}{2},j,k} + 9\phi_{i+\frac{1}{2},j,k} - \phi_{i+\frac{3}{2},j,k}}{16}.$$
(27)

The biased version of this operator (subscript *b*) can be applied in the vicinity of the boundaries at the bottom and top. Here, we show the biased stencil that can be applied for vertical interpolation near the bottom

10
$$\phi_{\underline{i,j}\,\underline{i,j,k}} \approx \underbrace{\frac{4xb}{i,j}}_{\underline{i,j}} \overline{\phi}_{\underline{i,j,k}}^{4zb} \equiv \frac{5\phi_{i,j,k-\frac{1}{2}} + 15\phi_{i,j,k+\frac{1}{2}} - 5\phi_{i,j,k+\frac{3}{2}} + \phi_{i,j,k+\frac{5}{2}}}{16}.$$
 (28)

Note that we only write down the bottom boundary for brevity.

The centered and biased fourth-order gradient operators are

$$\frac{\partial \phi}{\partial x} \bigg|_{\substack{i,j_{i,j,k} \\ \equiv}} \approx \delta^{4x}(\phi)_{\underline{i,j_{i,j,k}}} \\ \equiv \frac{\phi_{i-\frac{3}{2},j,k} - 27\phi_{i-\frac{1}{2},j,k} + 27\phi_{i+\frac{1}{2},j,k} - \phi_{i+\frac{3}{2},j,k}}{x_{i-\frac{3}{2}} - 27x_{i-\frac{1}{2}} + 27x_{i+\frac{1}{2}} - x_{i+\frac{3}{2}}},$$
(29)

15 and

20

$$\begin{aligned} \frac{\partial \phi}{\partial z} \bigg|_{\substack{i,ji,j,k \\ \to \infty}} &\approx \quad \delta \frac{4xb4zb}{\infty}(\phi)_{\underline{i,ji,j,k}} \\ &\equiv \quad \frac{-23\phi_{i,j,k-\frac{1}{2}} + 21\phi_{i,j,k+\frac{1}{2}} + 3\phi_{i,j,k+\frac{3}{2}} - \phi_{i,j,k+\frac{5}{2}}}{-23x_{i-\frac{1}{2}}z_{k-\frac{1}{2}}} + 21x_{i+\frac{1}{2}}z_{k+\frac{1}{2}} + 3x_{i+\frac{3}{2}}z_{k+\frac{3}{2}} - x_{i+\frac{5}{2}}z_{k}(3\theta) \\ &= \quad \frac{-23\phi_{i,j,k-\frac{1}{2}} + 21\phi_{i,j,k+\frac{1}{2}} + 3\phi_{i,j,k+\frac{3}{2}} - \phi_{i,j,k+\frac{5}{2}}}{-23x_{i-\frac{1}{2}}z_{k-\frac{1}{2}}} + 21x_{i+\frac{1}{2}}z_{k+\frac{1}{2}} + 3x_{i+\frac{3}{2}}z_{k+\frac{3}{2}} - x_{i+\frac{5}{2}}z_{k}(3\theta) \\ &= \quad \frac{-23\phi_{i,j,k-\frac{1}{2}} + 21\phi_{i,j,k+\frac{1}{2}} + 3\phi_{i,j,k+\frac{3}{2}} - \phi_{i,j,k+\frac{5}{2}}}{-23x_{i-\frac{1}{2}}z_{k-\frac{1}{2}}} + 21x_{i+\frac{1}{2}}z_{k+\frac{1}{2}} + 3x_{i+\frac{3}{2}}z_{k+\frac{3}{2}} - x_{i+\frac{5}{2}}z_{k}(3\theta) \\ &= \quad \frac{-23\phi_{i,j,k-\frac{1}{2}} + 21\phi_{i,j,k+\frac{1}{2}} + 3\phi_{i,j,k+\frac{3}{2}} - \phi_{i,j,k+\frac{5}{2}}}{-23x_{i-\frac{1}{2}}z_{k-\frac{1}{2}}} + 21x_{i+\frac{1}{2}}z_{k+\frac{1}{2}} + 3x_{i+\frac{3}{2}}z_{k+\frac{3}{2}} - x_{i+\frac{5}{2}}z_{k}(3\theta) \\ &= \quad \frac{-23\phi_{i,j,k-\frac{1}{2}} + 21\phi_{i,j,k+\frac{1}{2}} + 3\phi_{i,j,k+\frac{3}{2}} - \phi_{i,j,k+\frac{5}{2}} - 23x_{i-\frac{1}{2}}z_{k-\frac{1}{2}} + 21x_{i+\frac{1}{2}}z_{k+\frac{1}{2}} + 3x_{i+\frac{3}{2}}z_{k+\frac{3}{2}} - x_{i+\frac{5}{2}}z_{k}(3\theta) \\ &= \quad \frac{-23\phi_{i,j,k-\frac{1}{2}} + 21\phi_{i,j,k+\frac{1}{2}} + 3\phi_{i,j,k+\frac{1}{2}} - 2x_{i+\frac{5}{2}}z_{k+\frac{1}{2}} -$$

**3.5 Boundary conditions**

The lateral boundaries in MicroHH are periodic. The bottom and top boundary conditions can be formulated in their most general form as the Robin boundary condition

$$\left. a\phi_s + b\frac{\partial\phi}{\partial z} \right|_s = c,\tag{31}$$

with a, b and c as constants. This gives the Dirichlet boundary condition when a = 1, b = 0, and the Neumann boundary condition when a = 0, b = 1.

MicroHH makes use of ghost cells in order to avoid the need of biased schemes for single interpolation or gradient operators near the wall. The values at the ghost cells are derived making use of the boundary conditions following Morinishi et al. (1998). The ghost cells for the Dirichlet boundary conditions in the second-order accurate discretization are

$$\phi_{-\frac{1}{2}} = \frac{2c - \phi_{\frac{1}{2}}}{2c - \phi_{\frac{1}{2}}},\tag{32}$$

5 whereas those for the Neumann boundary condition are

$$\phi_{-\frac{1}{2}} = -c\left(-z_{-\frac{1}{2}} + z_{\frac{1}{2}}\right) + \phi_{\frac{1}{2}}.$$
(33)

In case of the fourth-order scheme, we have two ghost cells, and therefore a second boundary condition is required. Here, we set the third derivative equal to zero following (Morinishi et al., 1998). For the Dirichlet boundary condition we then acquire the following expressions for the ghost cells

10
$$\phi_{-\frac{1}{2}} = \frac{8c - 6\phi_{\frac{1}{2}} + \phi_{\frac{3}{2}}}{3},$$
 (34)

$$\phi_{-\frac{3}{2}} = \frac{8c - 6\phi_{\frac{1}{2}} + \phi_{\frac{3}{2}}}{2}, \tag{35}$$

whereas in case of a Neumann boundary condition we find

$$\phi_{-\frac{1}{2}} = -c \frac{z_{-\frac{3}{2}} - 27z_{-\frac{1}{2}} + 27z_{\frac{1}{2}} - z_{\frac{3}{2}}}{24} + \phi_{\frac{1}{2}}, \tag{36}$$

$$\phi_{-\frac{3}{2}} = -3c \frac{z_{-\frac{3}{2}} - 27z_{-\frac{1}{2}} + 27z_{\frac{1}{2}} - z_{\frac{3}{2}}}{24} + \phi_{\frac{3}{2}}.$$
(37)

**15 3.6 Advection**

20

25

o 1

We use the previously introduced notation to describe the more complex operators and expand them for illustration. The advection term is discretized in the flux form, where  $\phi$  is an arbitrary scalar located in the center of the grid cell. In the second-order case, this gives the following discretization:

The discretization of the advection of the velocity components (see Eqs. 5 and 7) involves extra interpolations as the following example illustrates:

$$\frac{\partial vu}{\partial x}\Big|_{\underline{i,j}\underline{i,j,k}} = \delta^{2x} \left(\overline{v}^{2y}\overline{u}^{2x}\right)_{\underline{i,j}\underline{i,j,k}} \\
= \frac{\overline{v}_{i+\frac{1}{2},j,k}^{2y}\overline{u}_{i+\frac{1}{2},j,k}^{2x} - \overline{v}_{i-\frac{1}{2},j,k}^{2y}\overline{u}_{i-\frac{1}{2},j,k}^{2x}}{x_{i+\frac{1}{2}} - x_{i-\frac{1}{2}}}.$$
(39)

In the standard fourth-order scheme, the scalar advection in flux form is represented by

$$\frac{\partial u\phi}{\partial x} \Big|_{\underline{i,ji,j,k}} \approx \delta^{4x} \left( u\overline{\phi}^{4x} \right)_{\underline{i,ji,j,k}} \\
= \left( u_{\underline{i-\frac{3}{2},ji-\frac{3}{2},j,k}} \overline{\phi}^{4x} \underbrace{i-\frac{3}{2},ji-\frac{3}{2},j,k}_{\underline{i-\frac{3}{2},ji-\frac{3}{2},j,k}} - 27u_{\underline{i-\frac{1}{2},ji-\frac{1}{2},j,k}} \overline{\phi}^{4x} \underbrace{i-\frac{1}{2},ji-\frac{1}{2},j,k}_{\underline{i-\frac{1}{2},ji+\frac{1}{2},j,k}} \overline{\phi}^{4x} \underbrace{i+\frac{1}{2},ji+\frac{1}{2},j,k}_{\underline{i+\frac{1}{2},ji+\frac{1}{2},j,k}} \overline{\phi}^{4x} \underbrace{i+\frac{1}{2},ji+\frac{1}{2},j,k}_{\underline{i+\frac{1}{2},ji+\frac{1}{2},j,k}} \overline{\phi}^{4x} \underbrace{i+\frac{3}{2},j,k}_{\underline{i+\frac{3}{2},j,k}} \overline{\phi}^{4x} \underbrace{i+\frac{3}{2},j,k}_{\underline{i+\frac{3}{2},j,k}} \overline{\phi}^{4x} \underbrace{i+\frac{3}{2},j,k}_{\underline{i+\frac{3}{2},j,k}} \right) \\
/ \left( x_{i-\frac{3}{2}} - 27x_{i-\frac{1}{2}} + 27x_{i+\frac{1}{2}} - x_{i+\frac{3}{2}} \right).$$
(40)

5

Hereafter, we assume that operator notation is clear and only expand it where necessary.

MicroHH has a fully kinetic energy-conserving fourth-order advection scheme (Morinishi et al., 1998) available. This The scheme is constructed by interpolation of two kinetic energy-conserving second-order schemes-discretizations to eliminate the second-order error as illustrated below

$$\quad \frac{\partial u\phi}{\partial x} \bigg|_{\substack{i,j\,i,j,k\\ \longrightarrow}} \approx \frac{9}{8} \delta^{2x} \left( u\overline{\phi}^{2x} \right)_{\underline{i,j\,i,j,k}} - \frac{1}{8} \delta^{2xL} \left( u\overline{\phi}^{2xL} \right)_{\underline{i,j\,i,j,k}}$$
(41)

to ensure that velocity variances are conserved under advection.

Velocity interpolations, such as those in Eq. 39, still need to be performed with fourth-order accuracy (Eq. 27) in order to be fourth-order accurate (see Morinishi et al. (1998) for details). The expression

$$\frac{\partial vu}{\partial x} \bigg|_{\underline{i,ji,j,k}} \approx \frac{9}{8} \delta^{2x} \left( \overline{v}^{4y} \overline{u}^{2x} \right)_{\underline{i,ji,j,k}} - \frac{1}{8} \delta^{2xL} \left( \overline{v}^{4y} \overline{u}^{2xL} \right)_{\underline{i,ji,j,k}}$$
(42)

15 includes, for instance, a combination of second- and fourth-order interpolations.

To increase the overall accuracy of the second-order advection operator, there is an option available to only increase the interpolation part to fourth order

$$\frac{\partial u\phi}{\partial x}\bigg|_{\substack{i,ji,j,k\\ \longrightarrow}} \approx \delta^{2x} \left(u\overline{\phi}^{4x}\right)_{\underline{i,ji,j,k}}.$$
(43)

**3.7 Diffusion**

.

We apply a discretization for diffusion that can be written as the divergence of a gradient, using the building blocks defined 20 earlier in this section. As this operator is identical in all directions, we present it in one direction only

$$\kappa_{\phi} \frac{\partial^2 \phi}{\partial x^2} \bigg|_{\underbrace{i,j\,i,j,k}_{\sim \sim \sim \sim}} \approx \kappa_{\phi} \delta^{2x} \left( \delta^{2x} \left( \phi \right) \right) \underbrace{i,j\,i,j,k}_{i,j\,i,j,k}, \tag{44}$$

$$\kappa_{\phi} \frac{\partial^2 \phi}{\partial x^2} \bigg|_{\underbrace{i,ji,j,k}_{\bullet \to \infty}} \approx \kappa_{\phi} \delta^{4x} \left( \delta^{4x} \left( \phi \right) \right) \underbrace{i,ji,j,k}_{\bullet \to \infty}.$$
(45)

On an equidistant grid, this provides the well-known second-order accurate operator for the second derivative

25
$$\kappa_{\phi}\delta^{2x}\left(\delta^{2x}\left(\phi\right)\right)_{\underline{i,j}\underline{i,j,k}} = \kappa_{\phi}\frac{\phi_{i-1,j,k} - 2\phi_{i,j,k} + \phi_{i+1,j,k}}{\left(\Delta x\right)^{2}},$$
(46)

where  $\Delta x$  is the uniform grid spacing.

Figure 1. Schematic of the diffusion discretization near the wall. The green node is the evaluation point at the center of the first cell above the wall, the red node is the stencil of the divergence operator, and yellow nodes show the stencils of the four gradient operators over which the divergence is evaluated. White nodes indicate the extent of the stencil.

For the fourth-order accurate operator, a seven-point stencil is used:

$$\kappa_{\phi} \delta^{4x} \left( \delta^{4x} \left( \phi \right) \right)_{\underline{i,ji,j,k}} = \frac{\kappa_{\phi}}{576 \left( \Delta x \right)^2} \left( \phi_{\underline{i-3,ji-3,j,k}} - 54 \phi_{\underline{i-2,ji-2,j,k}} + 783 \phi_{\underline{i-1,ji-1,j,k}} - 1460 \phi_{\underline{i,ji,j,k}} + 783 \phi_{\underline{i+1,ji+1,j,k}} - 54 \phi_{\underline{i+2,ji+2,j,k}} + \phi_{\underline{i+3,ji+3,j,k}} \right).$$

$$(47)$$

5 The seven point wide stencil and its properties has been discussed in detail in Castillo et al. (1995).

Whereas diffusion can be computed with fourth-order accuracy using a five-point stencil, we use a seven-point stencil as it extends naturally to non-uniform grids as explained in Castillo et al. (1995). The usage of a seven-point stencil requires special care near the walls. In Fig. 1 we show an example of how the second derivative in the vertical direction is computed for a scalar at the first model level (green node in Fig. 1). The calculation of the divergence (Fig. 1, red stencil) requires the gradient located

10 at the first face below the wall (lowest red node in Fig. 1), which can only be acquired using the biased gradient operator (Eq. 30 and yellow stencil connected to lowest red node in Fig. 1). The extent of the complete stencil near the wall (white nodes, Fig. 1) is thus six points, rather than seven.

**3.8 Pressure**

Eqs. 8 and 9 are solved following the method of Chorin (1968). This is a fractional step method that first computes intermediate values of the velocity components for the next time step, based on all right hand side terms of the momentum conservation equation Eq. 5

5
$$\underbrace{u_i^*}_{i,j,k} \Big|_{i,j,k}^{t+1} = u_i \Big|_{i,j,k}^t + \Delta t f_i \Big|_{i,j,k}^t,$$
 (48)

with the intermediate velocity components denoted with a tildean asterix.

. . .

The velocity values at the next time step can be computed as soon as the pressure is known, using

$$u_{i}|_{i,j,k}^{t+1} = \underbrace{u_{i}^{*}}_{i,j,k} \Big|_{i,j,k}^{t+1} - \Delta t \, \delta^{nx_{i}} \left(\frac{p}{\rho_{0}}\right) \Big|_{i,j,k}^{t}.$$
(49)

In order to compute the pressure, we multiply the previous equation with the reference density and take its gradient, arriving at

$$\quad \delta^{nx_{i}}(\rho_{0}u_{i})|_{i,j,k}^{t+1} = \left. \delta^{nx_{i}}\left(\rho_{0}\underline{u_{i}^{*}}\right)\right|_{i,j,k}^{t+1} \\ - \left. \Delta t \, \delta^{nx_{i}}\left[\rho_{0}\delta^{nx_{i}}\left(\frac{p}{\rho_{0}}\right)\right]\right|_{i,j,k}^{t},$$

$$(50)$$

where *n* indicates the spatial order, and the subscript *i* in superscript  $x_i$  indicates that  $\delta^{nx_i}$  is a divergence operator. The left hand side equals zero due to mass conservation at the next time step (Eq. 2). The resulting equation is the Poisson equation that is the discrete equivalent of Eq. 8. Rewriting this equation leads to

$$\quad \frac{\delta^{nx_i} \left(\rho_0 u_i^*\right)|_{i,j,k}^{t+1}}{\Delta t} \quad = \quad \delta^{nx_i} \left[\rho_0 \delta^{nx_i} \left(\frac{p}{\rho_0}\right)\right] \Big|_{i,j,k}^t.$$

$$(51)$$

To simplify the notation, we denote the left-hand side term as  $\psi$  and the  $p/\rho_0$  term on the right hand side as  $\pi$ . Solving a Poisson equation is a global operation. Because the computed fields are periodic in the horizontal directions on an equidistant grid, and a Poisson equation is linear, we can perform a Fourier transform in the two horizontal directions

$$\widehat{\psi}_{l,m,k} = -k_{*n}^2 \widehat{\pi}_{l,m,k} - l_{*n}^2 \widehat{\pi}_{l,m,k} + \delta^{nz} \left[ \rho_0 \delta^{nz} \left( \widehat{\pi} \right) \right]_{l,m,k},$$
(52)

20 where Fourier transformed variables are denoted with a hat, the spatial order of the operation with n, and the wave numbers in the two horizontal dimensions x and y are l and m respectively. Variables  $k_*^2$  and  $l_*^2$  are the squares of the modified wave numbers

$$-k_{*2}^2 \equiv 2\frac{\cos(k\Delta x)}{(\Delta x)^2} - \frac{2}{(\Delta x)^2}$$
(53)

and

$$25 -k_{*4}^{2} \equiv 2 \frac{\cos(3k\Delta x) - 54\cos(2k\Delta x) + 783\cos(k\Delta x)}{576(\Delta x)^{2}} - \frac{1460}{576(\Delta x)^{2}},$$
(54)

where the former is the modified wave number for the second-order accurate solver and the latter is the wave number for the fourth-order one. Note that the coeffecients correspond to those in Eqs. 46 and 47. Both expressions satisfy the limit  $\lim_{\Delta x\to 0} k_{*n}^2 = k^2$ , where *n* is the order of the scheme.

Solving Eq. 52 for  $\hat{\pi}$  requires solving a banded matrix , which for the vertical direction in which the walls are located. This matrix is tridiagonal for the second-order solver and hepta-diagonal for the fourth-order solver. For this, a standard Thomas algorithm (Thomas, 1949) is used. After the pressure is acquired, inverse Fourier transforms are applied and subsequently the pressure gradient term (see Eqs. 5 and 7) is computed for all three components of the velocity tendency. Note that the computation of the corrected velocity components does not require a boundary condition for pressure (see Vreman (2014) for details).

**10 3.9 Boundary conditions**

The lateral boundaries in MicroHH are periodic. The bottom and top boundary conditions can be formulated in their most general form as the Robin boundary condition

$$\left. a\phi_s + b \left. \frac{\partial \phi}{\partial z} \right|_s = c,$$

with a, b and c as constants. This gives the Dirichlet boundary condition when a = 1, b = 0, and the Neumann boundary 15 condition when a = 0, b = 1.

MicroHH makes use of ghost cells in order to avoid the need of biased schemes for single interpolation or gradient operators near the wall. The values at the ghost cells are derived making use of the boundary conditions following Morinishi et al. (1998). The ghost cells for the Dirichlet boundary conditions in the second-order accurate discretization are-

$$\phi_{-\frac{1}{2}} \equiv 2c - \phi_{\frac{1}{2}},$$

20 whereas those for the Neumann boundary condition are-

$$\underline{\phi_{-\frac{1}{2}}} = -c\left(-z_{-\frac{1}{2}} + z_{\frac{1}{2}}\right) + \phi_{\frac{1}{2}}.$$

In case of the fourth-order scheme, we have two ghost cells, and therefore a second boundary condition is required. Here, we set the third derivative equal to zero following (Morinishi et al., 1998). For the Dirichlet boundary condition we then acquire the following expressions for the ghost cells-

25
$$\underline{\phi_{-\frac{1}{2}}} \equiv \frac{8c - 6\phi_{\frac{1}{2}} + \phi_{\frac{3}{2}}}{3},$$

 $\underline{\phi_{-\frac{3}{2}}} \equiv \frac{8c - 6\phi_{\frac{1}{2}} + \phi_{\frac{3}{2}}}{3},$

whereas in case of a Neumann boundary condition we find

$$\frac{\phi_{-\frac{1}{2}}}{\phi_{-\frac{3}{2}}} \equiv \frac{-c\frac{z_{i-\frac{3}{2}} - 27z_{i-\frac{1}{2}} + 27z_{i+\frac{1}{2}} - z_{i+\frac{3}{2}}}{24} + \phi_{\frac{1}{2}},}{-3c\frac{z_{i-\frac{3}{2}} - 27z_{i-\frac{1}{2}} + 27z_{i+\frac{1}{2}} - z_{i+\frac{3}{2}}}{24} + \phi_{\frac{3}{2}}.}$$

**3.9 Thermodynamics**

5 MicroHH supports the potential ( $\theta$ ) and liquid water potential ( $\theta_1$ ) temperature as thermodynamic variables (Sect. 2.5). The dry ( $\theta$ ) and moist ( $\theta_1$ ) thermodynamics are related through the use of a total specific humidity  $q_t$ , which is defined as the sum of the water vapour specific humidity ( $q_v$ ) and the cloud liquid water specific humidity ( $q_1$ ). In the absence of liquid water,  $\theta_1 = \theta$ , in the presence of liquid water, the liquid water potential temperature is approximated as (Betts, 1973)

$$\theta_{\rm l} \approx \theta - \frac{L_{\rm v}}{c_{\rm p} \Pi} q_{\rm l},\tag{55}$$

10 where  $L_v$  is the latent heat of vaporization,  $c_p$  the specific heat of dry air at constant pressure, and  $\Pi$  is the Exner function

$$\Pi = \left(\frac{p}{p_{00}}\right)^{R_{\rm d}/c_{\rm p}},\tag{56}$$

where p is the hydrostatic pressure,  $p_{00}$  a constant reference pressure, and  $R_d$  the gas constant for dry air. The cloud liquid water content is calculated as

$$q_1 = \max(0, q_t - q_s), \tag{57}$$

15 where  $q_s$  is the saturation specific humidity

20

$$q_{\rm s} = \frac{\epsilon \, e_{\rm s}}{p - (1 - \epsilon) \, e_{\rm s}},\tag{58}$$

with  $\epsilon$  the ratio between the gas constant for dry air and the gas constant for water vapour  $(R_d/R_v)$ , and  $e_s$  the saturation vapor pressure. The latter is approximated using a 10th order Taylor expansion at T = 0 degree Celsius of the Arden Buck equation (Buck, 1981).  $q_l$  is adjusted iteratively to arrive at a consistent state where  $q_v = q_s$ . Finally, the virtual potential temperature (Eq. 5) is defined in MicroHH as

$$\theta_{\rm v} \equiv \theta \left( 1 - \left[ 1 - \frac{R_{\rm v}}{R_{\rm d}} \right] q_{\rm t} - \frac{R_{\rm v}}{R_{\rm d}} q_{\rm l} \right) \tag{59}$$

The base state pressure and density are calculated assuming a hydrostatic equilibrium:  $dp_0 = -\rho_0 g dz$ , with the density defined as  $\rho_0 = p_0/(R_d \Pi \theta_{v0})$ . Integration with height results in

$$p_{0;k+1} = p_{0;k} \exp\left(\frac{-g(z_{k+1} - z_k)}{R_d \,\Pi \,\theta_{v0}}\right) \tag{60}$$

where θv0 is the average virtual potential temperature between zk and zk+1. This equation is applied from a given surface
pressure to the model top, alternating the calculations at the full and half model levels. That is, given the full thermodynamic state (pressure and density) at a full level k, the thermodynamic state can be advanced from the half level k - 1/2 to k + 1/2. Using the newly calculated state at k + 1/2, pressure and density at k + 1 can be calculated.

The base state density ρ0 that is used in the dynamical core (Sect. 2) is calculated using the initial virtual potential temperature profile, and is not updated during the experiment. The density and hydrostatic pressure used in the moist thermodynamics
10 can optionally be updated every time step, following the same procedure as explained in Boing (2014).

**3.10 Rotation**

The effects of a rotating reference frame on an f-plane can be included through the Coriolis force. MicroHH can run on an f-plane, where the related tendencies of The acceleration due to the Coriolis force  $F_{i,cor}$  is computed for the two horizontal velocity components are calculated as (index 1 and 2 in Eqs. 5 and 7) as

15
$$F_{1,cor}\Big|_{i,j,k,F_{cor}i,j,k} = f_0 v_{i,j,k},$$
 (61)

$$\frac{F_{2,\text{cor}}}{F_{2,\text{cor}}}\Big|_{i,j,k,F_{\text{cor}}i,j,k} = -f_0 u_{i,j,k}, \tag{62}$$

with  $f_0$  as Coriolis parameter specified by the user.

**4 Physical parameterizations**

**4.1 Subfilter-scale model for large-eddy simulation**

20 With the governing equations described in Sect. 2 it is possible to resolve the flow down to the scales where molecular viscosity acts. In many applications, however, such simulations are too costly. In that case, one may opt for large-eddy simulation (LES), where filtered equations are used to describe the largest scales of the flow, and the subfilter-scale motions are modeled. The LES implementation in MicroHH assumes very high Reynolds numbers in which the molecular viscosity is neglected. Filtering of the anelastic conservation of momentum equation (Eq. 5), with a tilde applied to denote filtered variables, leads to

$$\quad \frac{\partial \widetilde{u}_{i}}{\partial t} = -\frac{1}{\rho_{0}} \frac{\partial \rho_{0} \widetilde{u}_{i} \widetilde{u}_{j}}{\partial x_{j}} - \frac{\partial \pi}{\partial x_{i}} - \frac{1}{\rho_{0}} \frac{\partial \rho_{0} \tau_{ij}}{\partial x_{j}} + \delta_{i3} g \frac{\widetilde{\theta}'_{v}}{\theta_{v0}} + F_{i}.$$
(63)

In this equation, a tensor  $\tau_{ij}$  is defined as

$$\tau_{ij} \equiv \widetilde{u_i u_j} - \widetilde{u}_i \widetilde{u}_j - \frac{1}{3} \left( \widetilde{u_i u_i} - \widetilde{u}_i \widetilde{u}_i \right).$$
(64)

This is the anisotropic subfilter-scale kinematic momentum flux tensor. The isotropic part of the full momentum flux tensor has been added to the pressure, providing the modified pressure

$$\quad \pi \equiv \frac{\widetilde{p}'}{\rho_0} + \frac{1}{3} \left( \widetilde{u_i u_i} - \widetilde{u}_i \widetilde{u}_i \right). \tag{65}$$

As  $\tau_{ij}$  contains the filtered product of unfiltered velocity components, this quantity needs to be parameterized. MicroHH uses the Smagorinsky-Lilly (Lilly, 1968) model, in which  $\tau_{ij}$  is modeled as

$$\tau_{ij} = -K_m \left( \frac{\partial \widetilde{u}_i}{\partial x_j} + \frac{\partial \widetilde{u}_j}{\partial x_i} \right), \tag{66}$$

with  $K_m$  interpreted as the subfilter eddy-diffusivity. This quantity is modeled as

10
$$K_m = \lambda^2 \underline{2} S_{\underline{ij}} \underline{Sij}^{\frac{1}{2}} \left( 1 - \frac{g}{\theta_{v0}} \frac{\partial \widetilde{\theta_v}}{\partial z}}{Pr_t S^2} \right)^{\frac{1}{2}},$$
 (67)

and is proportional to the magnitude of S of  $S \equiv (2S_{ij}S_{ij})^{\frac{1}{2}}$  of the strain tensor  $S_{ij}$ , which is defined as

$$S_{ij} \equiv \frac{1}{2} \left( \frac{\partial \widetilde{u}_i}{\partial x_j} + \frac{\partial \widetilde{u}_j}{\partial x_i} \right).$$
(68)

The subfilter eddy diffusivity thus takes into account the local stratification N2 = (g/θv0)/(∂θ̃v/∂z) and the turbulent Prandtl number Prt. The latter is set to 1/3 
[revised manuscript text omitted]

25
$$\frac{F_{1,p;ls}}{F_{2,p;ls}} \Big|_{\substack{i,j,k,F_{p;ls}i,j,k \\ i,j,k,F_{p;ls}i,j,k \\ i,j,k,F$$

where  $u_{g;k}$  and  $v_{g;k}$  as are user-specified vertical profiles of geostrophic wind components.

**4.3.2 Large-scale sources and sinks**

Large-scale sources and sinks, representing for instance large-scale advection advection or radiative cooling, can be prescribed for each variable separately. The user has to provide vertical profiles of large-scale tendencies  $S_{\phi;ls}$  sources and sinks  $S_{\phi;ls}$  that are added to the total tendencies.

**5 4.3.3 Large-scale vertical velocity**

A second method of introducing large-scale thermodynamic effects is through the inclusion of a large-scale vertical velocity. In this case, each scalar gets an additional tendency term source term  $S_{\phi,w,ls}$  of the form

$$S_{\underbrace{\phi, w, ls}}_{\underbrace{i, j, k, lsi, j, k}} = -w_{\underline{ls; kls; k}} \delta^{2x} \left( \langle \phi \rangle_k \right), \tag{92}$$

where  $w_{ls;k}$  is a user-specified vertical profile of large-scale vertical velocity,  $\langle \phi \rangle_k$  is the horizontally-averaged vertical profile 10 at height  $z_k$  for scalar  $\phi$ . The tendency term is not applied to the momentum variables.

**4.4 Buffer layer**

MicroHH has the option to damp gravity waves in the top of the simulation domain in a so-called buffer layer. The tendency source term  $S_{\phi,buf}$  associated with the damping at grid cell i, j, k is calculated for an arbitrary variable  $\phi$  as

$$\underbrace{S_{\phi,\text{buf}}}_{\sim\sim\sim\sim} \left| \underbrace{i,j,k,\text{buf}_{i,j,k}}_{\sim\sim\sim} = \
[revised manuscript text omitted]